# tauFisher predicts circadian time from a single sample of bulk and single-cell pseudobulk transcriptomic data

Junyan Duan [1,2,12], Michelle N. Ngo [1,2,12], Satya Swaroop Karri [3], Lam C. Tsoi[4,5,6,7], Johann E. Gudjonsson [4,7], Babak Shahbaba [1,8] ✉, John Lowengrub [1,9,10] ✉ & Bogi Andersen [1,3,11] ✉

As the circadian clock regulates fundamental biological processes, disrupted clocks are often observed in patients and diseased tissues. Determining the circadian time of the patient or the tissue of focus is essential in circadian medicine and research. Here we present tauFisher, a computational pipeline that accurately predicts circadian time from a single transcriptomic sample by finding correlations between rhythmic genes within the sample. We demonstrate tauFisher's performance in adding timestamps to both bulk and single-cell transcriptomic samples collected from multiple tissue types and experimental settings. Application of tauFisher at a cell-type level in a single-cell RNAseq dataset collected from mouse dermal skin implies that greater circadian phase heterogeneity may explain the dampened rhythm of collective core clock gene expression in dermal immune cells compared to dermal fibroblasts. Given its robustness and generalizability across assay platforms, experimental setups, and tissue types, as well as its potential application in single-cell RNAseq data analysis, tauFisher is a promising tool that facilitates circadian medicine and research.

Organisms have evolved intrinsic circadian clocks that help them anticipate and adjust to environmental changes caused by the 24-hour rotation of the Earth[1,2]. The mammalian circadian clock is a biochemical oscillator powered by transcription-translation loops consisting of a positive arm and a negative arm[1–3]. In the positive arm, BMAL1 and CLOCK promote the expression of clock-controlled genes, including the negative arm factors PER and CRY. PER and CRY inhibit the activating effect of BMAL1-CLOCK, leading to 24-hour oscillations.

In mammals, the suprachiasmatic nucleus (SCN) of the hypothalamus is the central pacemaker that coordinates and synchronizes circadian rhythms in peripheral tissues through neuronal and hormonal signals[4]. Besides signals from the SCN, environmental signals such as temperature[4], feeding[5,6], and direct light[7] can selectively set peripheral clocks, sometimes causing asynchrony between the central and peripheral clocks. Epidemiological studies of shift workers and chronically jet-lagged individuals show correlations between circadian disruption and cardiovascular diseases[8], mental health disorders[9],

[1]Center for Complex Biological Systems, University of California Irvine, Irvine, CA, USA. [2]The NSF-Simons Center for Multiscale Cell Fate Research, University of California Irvine, Irvine, CA, USA. [3]Department of Biological Chemistry, School of Medicine, University of California Irvine, Irvine, CA, USA. [4]Department of Dermatology, Michigan Medicine, University of Michigan, Ann Arbor, MI, USA. [5]Department of Computational Medicine and Bioinformatics, Michigan Medicine, University of Michigan, Ann Arbor, MI, USA. [6]Department of Biostatistics, School of Public Health, University of Michigan, Ann Arbor, MI, USA. [7]Mary H Weiser Food Allergy Center, University of Michigan, Ann Arbor, MI, USA. [8]Department of Statistics, University of California Irvine, Irvine, CA, USA. [9]Department of Mathematics, University of California, Irvine, CA, USA. [10]Department of Biomedical Engineering, University of California Irvine, Irvine, CA, USA. [11]Department of Medicine, Division of Endocrinology, School of Medicine, University of California Irvine, Irvine, CA, USA. [12]These authors contributed equally: Junyan Duan, Michelle N. Ngo. ✉e-mail: babaks@uci.edu; jlowengr@uci.edu; bogi@uci.edu

metabolic diseases[10–12], as well as cancer in various organs[13,14], including skin[15,16], breast[17,18], and prostate[19,20].

The goal of the nascent field of circadian medicine is to consider circadian rhythm and its disruption in patient care. As the rhythm of a patient or diseased tissue is not necessarily synchronized with the external light-dark cycle, an important challenge in circadian medicine is to determine the internal circadian time of the patient or the tissue of focus. Such information can determine the optimal time of treatment and identify conditions that might benefit from restoring circadian functions[21,22]. Current methods of circadian time determination for a patient include the dim-light melatonin-onset assay[23], as well as circadian rhythm inference from body temperature[24], or cortisol levels in biofluids[25].

Additionally, determining the circadian time of a sample is important for research. With the explosion of bulk and single-cell transcriptomics datasets, there is a growing effort to integrate and compare such datasets. As about 10% of the transcriptome has diurnal expression patterns, analyzing such datasets without their timestamps may lead to inconsistent observations that are dependent on the time of sample collection. Hence, there is a need to develop a method that can determine the circadian time of such datasets.

Several groups have developed methods to infer the circadian time of a sample (organism, organ, or tissue) based on transcriptomic data. CYCLOPS[26,27] uses an autoencoder neural network to infer circadian phases by ordering the data collected from the entire periodic cycle. ZeitZeiger[28] identifies useful features (genes) for prediction, scales the feature expression over time, applies sparse principal component analysis, and predicts according to maximum likelihood estimation. BIO_CLOCK[29] uses supervised deep neural networks with coupled sine and cosine output units. TimeSignatR[30] applies within-subject renormalization and an elastic net predictor, making it generalizable between transcriptomic data from different assay platforms. More recently, a Bayesian variational inference approach called Tempo[31] was designed to predict the circadian phase in single-cell transcriptomics and to quantify estimation uncertainty.

Among the methods mentioned above, CYCLOPS, ZeitZeiger, BIO_CLOCK, and TimeSignatR can infer circadian time from bulk transcriptomic data and are generalizable for different tissues. But, they have limitations. CYCLOPS outputs the relative ordering, instead of timestamps, of samples, and requires reconstruction to incorporate every new sample as it does not require prior training data. ZeitZeiger frequently runs into linear dependency issues, needs to be retrained before each prediction, and is not generalizable between transcriptomic platforms. BIO_CLOCK does not require re-training for each prediction but is not time-efficient. TimeSignatR performs well if there are two test samples and it achieves its best performance when the two samples are 12 hours apart.

Here, we present tauFisher, a pipeline that can accurately predict circadian time from a single transcriptomic data irrespective of the assay platform. tauFisher improves on previous methods in several ways: (1) it does not require the training data to be a complete time series; (2) the within-sample normalization step allows tauFisher to give an accurate prediction from just one sample; (3) since tauFisher only needs a few features to make accurate predictions, training and testing are computationally efficient; (4) tauFisher is platform agnostic and users only need to train the predictor once and can use the same predictor to make predictions for external datasets of the same tissue, regardless of the platform; and (5) tauFisher trained on bulk sequencing data is able to accurately predict the circadian time of single-cell RNA sequencing (scRNAseq) pseudobulk data, and it can be used to investigate circadian phase heterogeneity in different cell types.

We collected a time series of scRNAseq data from mouse dermis in this study and found that most of the rhythmic processes are metabolism-related in dermal fibroblasts, whereas almost all rhythmic processes are related to immune responses in dermal immune cells. Additionally, we found that the amplitude of the collective rhythm is dampened in dermal immune cells compared to dermal fibroblasts. Incorporating tauFisher with bootstrapping revealed that circadian phase heterogeneity contributes to the dampened collective rhythm as well as fewer rhythmic genes found in dermal immune cells.

## Results

### Overview of tauFisher

tauFisher is an assay platform-agnostic method that predicts circadian time from a single transcriptomic sample. The training part of the pipeline, which requires a time series of transcriptomic data, consists of five main steps: (1) identifying diurnal genes with a period length of 24 hours, (2) curve fitting using functional data analysis to fill in the missing time points and to decrease noise in the training data, (3) within-sample normalization by calculating and scaling the difference in expression for each pair of predictor genes, (4) linearly transforming the scaled differences using principal component analysis, and (5) fitting a multinomial regression on the first two principal components (Fig. 1, Methods).

For testing, a transcriptomic sample without a time label is trimmed to include only the predictor genes identified in the training data. After the within-sample normalization step, the test sample is projected to the principal component space, and multinomial regression is performed to predict the time of the test sample (Fig. 1, Methods).

### tauFisher achieves high accuracy when trained and tested on bulk-level transcriptomic data

To assess the robustness and accuracy of tauFisher in predicting circadian time from a single sample of transcriptomic data, we applied tauFisher to a diverse set of data collected from different species, tissues, and assay platforms (Table 1).

For each benchmark dataset, we generated 100 random train and test partitions (without replacement) of the samples. In each partition, we used 80% of the samples for training and 20% for testing. We compared tauFisher to the current state-of-the-art methods: ZeitZeiger[28] and TimeSignatR[30].

We define a prediction within two hours of the true time to be correct. Using other time ranges to define correctness minimally changes the benchmark outcome (Supplementary Table 1).

For eleven out of the twelve benchmarking datasets, tauFisher achieved higher accuracy when using predictor genes found by JTK_Cycle[32] instead of Lomb-Scargle[33]. For six out of the ten transcriptomic datasets collected from mice, tauFisher achieved equal or higher 2-hour accuracy using one test sample than TimeSignatR using two test samples that are 12 hours apart. tauFisher achieved lower but comparable accuracy (difference < 10%) when compared to Time-SignatR in two of the remaining four mouse datasets. For the two human blood datasets, TimeSignatR, using two test samples, outperformed ZeitZeiger and tauFisher, highlighting the importance and effectiveness of using two test samples to address human variability in circadian phase predictions (Fig. 2, Supplementary Table 2). ZeitZeiger could not predict the time for several iterations due to linearly dependent basis vectors. Interestingly, whether ZeitZeiger ran into linear dependency issues appeared to depend on the assay methods, as it ran successfully for most of the microarray data but failed to predict the time for all 100 iterations in the bulk RNAseq datasets collected from mouse kidney, liver, brainstem, and cerebellum (Fig. 2, Supplementary Table 2).

### tauFisher accurately predicts the circadian time for cross-platform bulk transcriptomic data

Since tauFisher gives accurate circadian time prediction for bulk transcriptomic data collected from various platforms, we examined its performance when trained and tested on datasets generated from

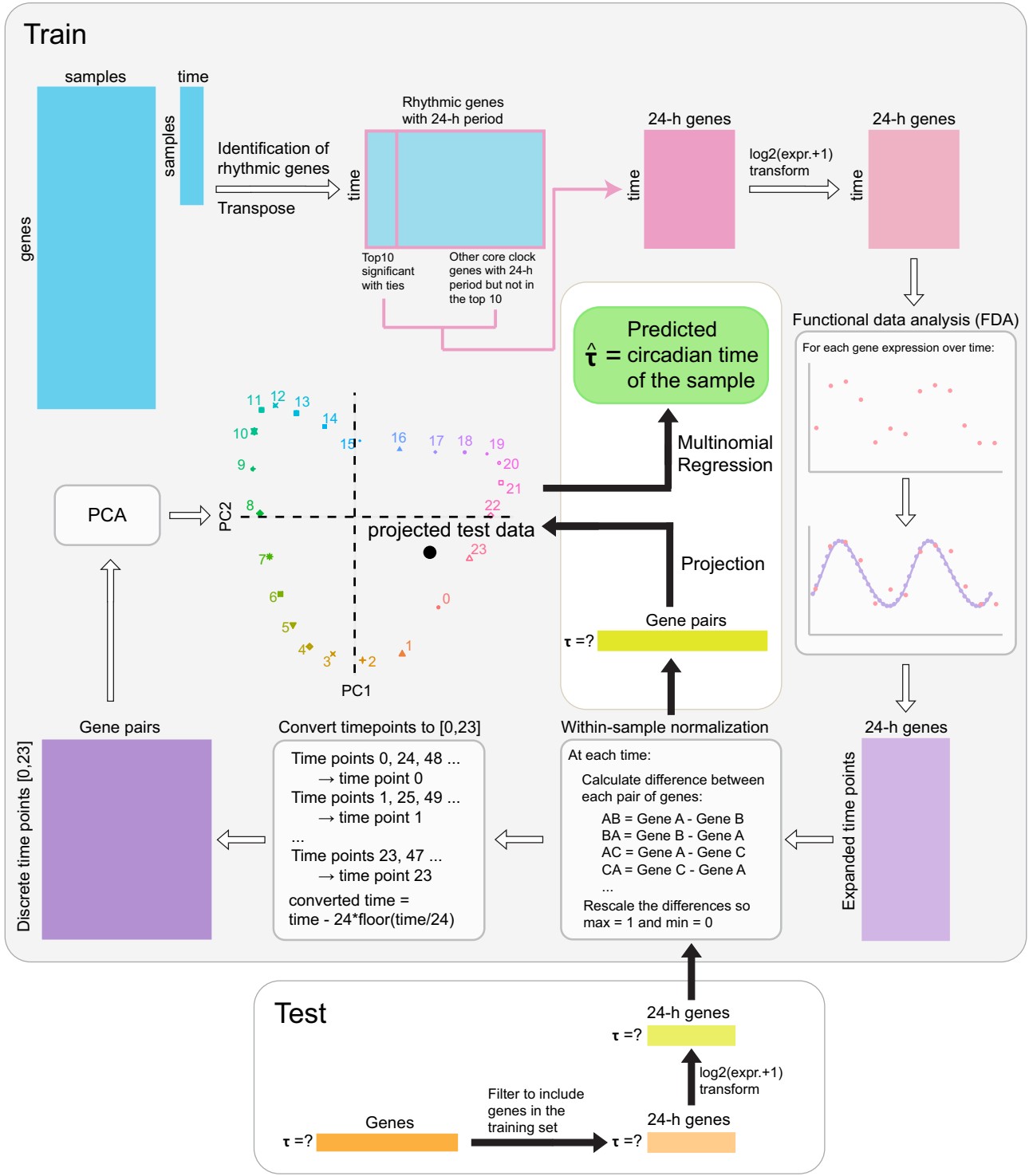

**Fig. 1 | The tauFisher pipeline involves multiple steps.** Key steps of the tauFisher pipeline include the identification of rhythmic genes using MetaCycle, functional data analysis, within-sample normalization, linear transformation, and multinomial regression.

different platforms. We used rhythmic genes identified by JTK_Cycle in the tauFisher pipeline since they gave more accurate predictions in the within-platform benchmark.

We trained tauFisher on GSE38622[34], a microarray dataset collected from mouse dorsal skin every four hours for 48 hours under a regular 12:12 light-dark cycle (zeitgeber time [ZT] 2, 6, 10, ...). The test dataset is from GSE83855[6], a bulk RNAseq dataset collected every four hours for 28 hours under a 12:12 light-dark cycle (ZT0, 4, 8, ...) from mouse dorsal skin in a time-restricted feeding study. Since the time of

feeding influences tissue's circadian clock[6,22], we only selected the ad libidum control condition for this testing so that the time labels best represent the internal time.

Eighteen genes were selected to be predictor features. Though the training and test datasets are not on the same scale and were collected at different time points, their overall rhythmic patterns agree with each other (Fig. 3a, Supplementary Fig. 1a). For six of the eight tests, tauFisher predicted a circadian time that is within the 2-hour range from the actual time label, giving an accuracy of 0.75 and a RMSE

**Table 1 | Datasets from different species, tissues, and assay platforms were used to benchmark tauFisher's ability to predict circadian time**

| Data | Year | GEO | Species | Tissue | Platform | Sampling Frequency | Time Course Duration |
|---|---|---|---|---|---|---|---|
| Zhang R et al.[56] | 2014 | GSE54650 | *Mus musculus* | Kidney | Affymetrix Mouse Gene 1.0 ST Array | 2h | 48h |
| Zhang R et al.[56] | 2014 | GSE54650 | *Mus musculus* | Liver | Affymetrix Mouse Gene 1.0 ST Array | 2h | 48h |
| Zhang R et al.[56] | 2014 | GSE54650 | *Mus musculus* | Brainstem | Affymetrix Mouse Gene 1.0 ST Array | 2h | 48h |
| Zhang R et al.[56] | 2014 | GSE54650 | *Mus musculus* | Cerebellum | Affymetrix Mouse Gene 1.0 ST Array | 2h | 48h |
| Zhang R et al.[56] | 2014 | GSE54651 | *Mus musculus* | Kidney | Illumina HiSeq 2000 | 6h | 48h |
| Zhang R et al.[56] | 2014 | GSE54651 | *Mus musculus* | Liver | Illumina HiSeq 2000 | 6h | 48h |
| Zhang R et al.[56] | 2014 | GSE54651 | *Mus musculus* | Brainstem | Illumina HiSeq 2000 | 6h | 48h |
| Zhang R et al.[56] | 2014 | GSE54651 | *Mus musculus* | Cerebellum | Illumina HiSeq 2000 | 6h | 48h |
| Arnardottir ES et al.[64] | 2014 | GSE56931 | *Homo sapiens* | Blood | Custom Affymetrix Microarray | 4h | 72h |
| Braun R et al.[30] | 2018 | GSE113883 | *Homo sapiens* | Blood | Illumina NextSeq 500 | 2h | 28h |
| Geyfman M et al.[34] | 2012 | GSE38622 | *Mus musculus* | Skin | Affymetrix Mouse Gene 1.0 ST Array | 4h | 48h |
| Tognini P et al.[36] | 2020 | GSE157077 | *Mus musculus* | SCN | Illumina HiSeq 4000 | 4h | 24h |

of 4.704 (Fig. 3b). This example demonstrates tauFisher's ability to predict circadian time across bulk transcriptomics platforms.

After validating tauFisher's performance on cross-platform, bulk-level, transcriptomic datasets collected from healthy/control mouse skin, we also tested it in disturbed systems. In the test groups of the time-restricted feeding study, food was only available to mice from ZT5 to ZT9 or ZT0 to ZT4, whereas mice usually feed during early nights (ZT12-ZT16)[6]. Skin collected from these two time-restricted feeding schedules showed disturbed circadian patterns with greatly attenuated amplitude and altered peaking times that are not directly correlated with the feeding times[6]. As the system is disturbed, the sample collection time no longer represents the internal circadian time of the tissue as it does in healthy tissue (training data). Consistent with the biological observations, tauFisher trained on control skin microarray data predicted time labels that are away from the test sample collection time, reflecting a disturbed system and the predictions are not coupled with time-restricted feeding schedules (Supplementary Fig. 2a). tauFisher's prediction when trained on control/healthy samples, however, can only tell whether the test system is disturbed or not, and does not provide a measurement of how much the system is disturbed.

We also trained the tested tauFisher within the disturbed systems. Within each of the two time-restricted feeding schedules, we performed leave-one-out cross-validation by reserving each sample for testing and using the remaining samples for training. tauFisher produced high accuracy (feeding ZT5-ZT9: accuracy = 0.875; feeding ZT0-ZT4: accuracy = 1) and low RMSE (feeding ZT5-ZT9: RMSE = 2.236; feeding ZT0-ZT4: RMSE = 1.061) for both disturbed systems (Supplementary Fig. 2b). The fact that tauFisher trained on samples collected from a disturbed system can add time labels to samples from the same disturbed system suggests that robust correlations between diurnal genes still exist in the disturbed system, and such relationships are different from the ones in the control/healthy individuals.

**tauFisher trained on bulk RNAseq data and microarray data accurately predicts the circadian time of scRNAseq samples**

tauFisher's ability to predict circadian time is not limited to cross-platform bulk-level transcriptomic datasets. It can add circadian timestamps to scRNAseq samples as well. In particular, tauFisher only needs to be trained on a time series of bulk-level transcriptomic data, which is more abundant and cheaper to collect than a scRNAseq data time series.

Since most published scRNAseq datasets do not have time labels, the selection of datasets for testing was limited. Here we tested tauFisher on scRNAseq data collected from the mouse SCN[35] and mouse dermal skin (collected in this study).

GSE117295[35] includes twelve single-cell SCN samples collected from circadian time (CT) 14 to 58 every four hours (CT14, 18, 22, ...) under constant darkness, and one light-stimulated SCN sample. Since light immediately induces differential expression of rhythmic genes[35], only the samples from the control experiment were used for the benchmark. For each of the twelve samples, a pseudobulk dataset was generated for testing (Methods). For training, we chose GSE157077[36], a time series of bulk RNAseq data collected from the mouse SCN every four hours under a regular 12:12 light-dark cycle starting at ZT0. Since each time point in the training dataset contains three replicates, instead of averaging them, we concatenated the replicates so that the input training data spans 72 hours.

Twenty genes from the training data passed the feature selection criteria. These genes display robust rhythms in both the training data and the test pseudobulk data (Fig. 3c, Supplementary Fig. 1b). The raw input test data appeared to be noisier as it was not normalized by the total number of reads in each sample. tauFisher does not require the data to be preprocessed before input into the pipeline, as within-sample normalization is an intermediary step.

In ten out of the twelve tests, tauFisher predicted a time that is within 2-hour of the labeled time, resulting in a high accuracy of 0.833 and a low RMSE of 1.936 (Fig. 3d, Supplementary Fig. 3a). Although neither TimeSignatR nor ZeitZeiger claims to be able to add time labels to scRNAseq data, we still tested their performance. tauFisher outperformed TimeSignatR and ZeitZeiger in both accuracy and RMSE (Fig. 3d, e).

To ensure that tauFisher's performance on scRNAseq data is consistent in peripheral clocks, we performed scRNAseq on adult wild-type C57BL/6J mouse dorsal dermis every four hours for 72 hours under 12:12 light-dark cycle. The pseudobulk matrices for the 18 samples were computed directly from the unprocessed data. We trained tauFisher on GSE38622[34], a time series of skin microarray data. Because two of the rhythmic genes, *A630005I04Rik* and *Ivl*, are not present in the pseudobulk data, only 16 features were selected in the tauFisher pipeline in this test (Fig. 3f, Supplementary Fig. 1c).

Although the input test data, the unnormalized pseudobulk data, appear to be noisy, tauFisher successfully predicts circadian times for the 18 samples thanks to the within-sample normalization step in the tauFisher pipeline. In 14 out of the 18 tests, tauFisher predicted circadian time within 2 hours of the labeled time, giving a high accuracy of 0.778 and a low RMSE of 2.198 (Fig. 3g, Supplementary Fig. 3b).

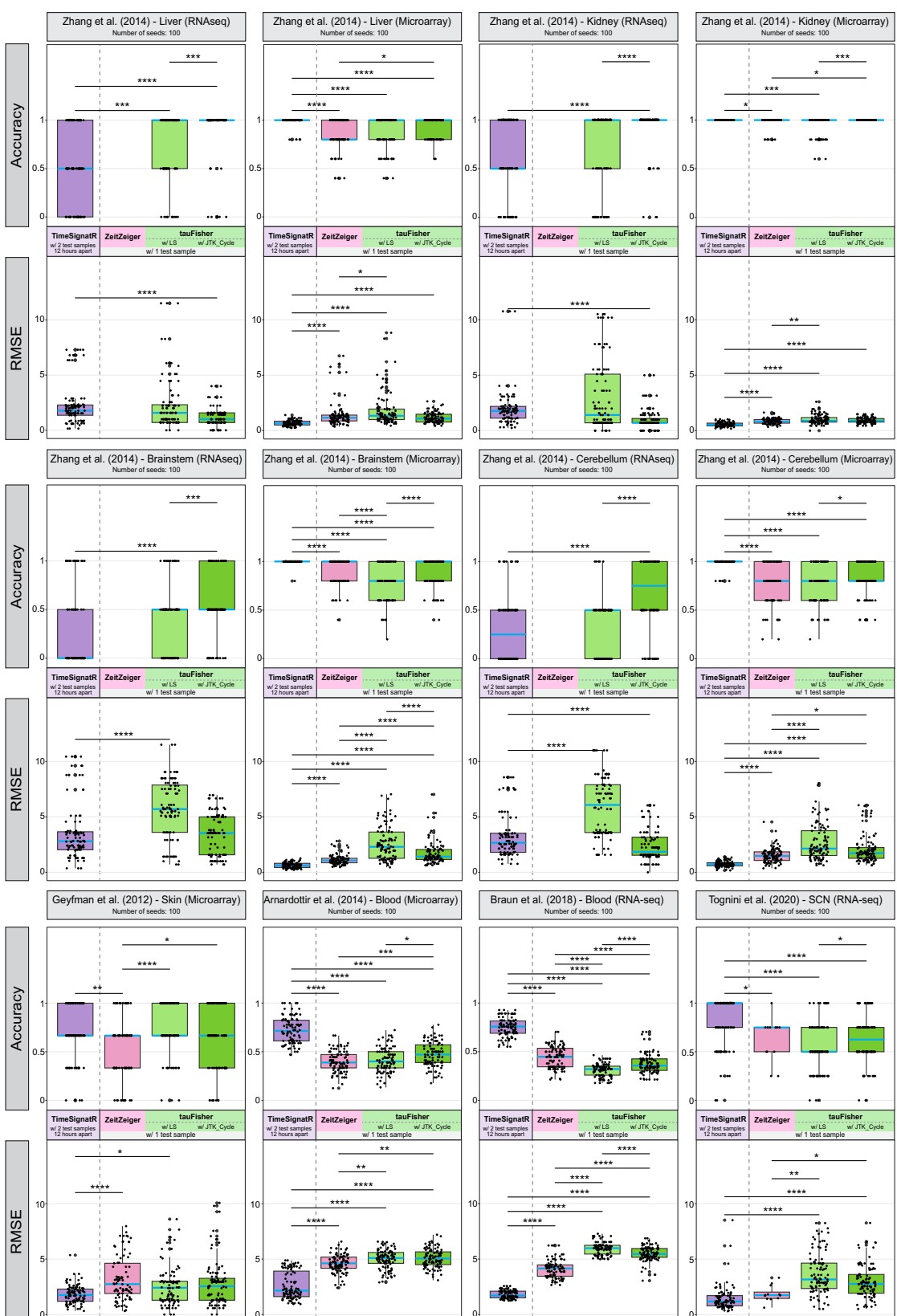

**Fig. 2 | tauFisher requires only one test sample and performs well in both accuracy and RMSE for transcriptomic data collected from various organs and assay platforms.** The blue line inside each box indicates the median. The bounds of the box represent the first and third quartiles. The upper and lower whiskers extend to the largest and smallest value within 1.5 times the inter-quartile range, respectively. NAs are excluded from the plot. RMSE: root mean square error. *: *p*-value ≤ 0.05, **: *p*-value ≤ 0.01, ***: *p*-value ≤ 0.001, ****: *p*-value ≤ 0.0001. *P*-values are determined using the Wilcoxon rank-sum test and adjusted using Bonferroni correction. For each dataset, *n* = 100 randomly generated training-testing partitions. Source data and exact *p*-values are provided as a Source Data file.

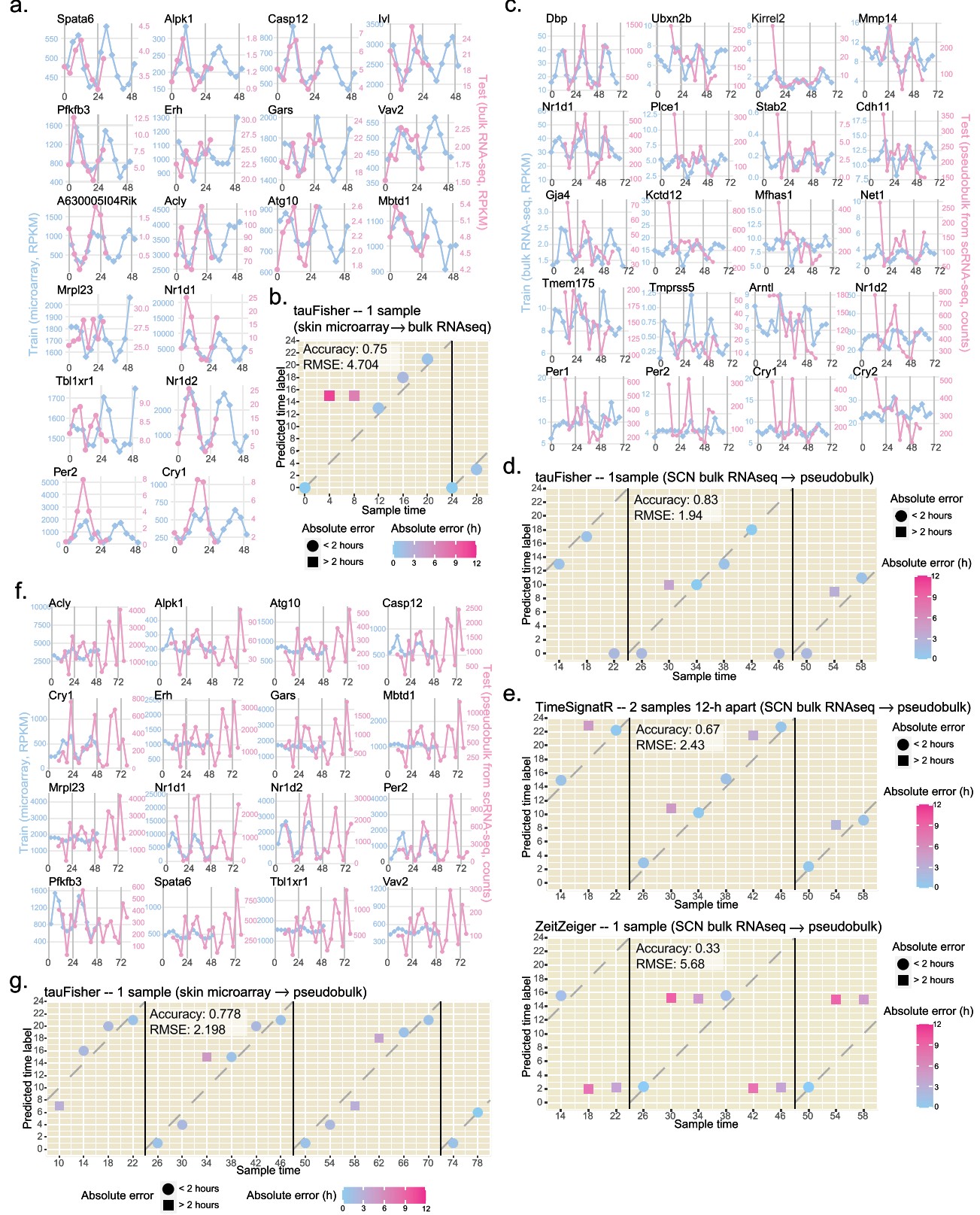

In sum, we have demonstrated that tauFisher trained on bulk-level transcriptomic data, either bulk RNAseq or microarray data, can accurately predict the circadian time for scRNAseq data, making it particularly useful for expanding the current scRNAseq database for circadian studies by adding time labels to existing scRNAseq data.

## Collective circadian rhythms are dampened in dermal immune cells compared to dermal fibroblasts

Due to the frequency of sequencing dropouts of clock genes in scRNAseq data, investigating the circadian clock within each cell is not yet achievable. To overcome this limitation, previous studies have used pseudobulk approaches to investigate the clock in scRNAseq data[35].

**Fig. 3 | tauFisher accurately predicts circadian time when the training and test data are from different assay methods.** tauFisher trained on mouse skin microarray data can predict circadian time for skin bulk RNAseq data. **a** Overlay of the predictor gene expression in GSE38622 (training) and GSE83855 (test). **b** Prediction outcomes from tauFisher. tauFisher trained on mouse SCN bulk RNAseq data can predict circadian time of pseudobulk data generated from mouse SCN scRNAseq data. **c** Overlay of the predictor gene expression in GSE157077 (training) and GSE117295 (test). **d** Prediction outcomes from tauFisher. **e** Prediction outcomes

from TimeSignatR and ZeitZeiger when trained on mouse SCN bulk RNAseq and tested on mouse SCN scRNAseq pseudobulk. tauFisher trained on mouse skin microarray data can predict circadian time of pseudobulk data generated from dermis scRNAseq data. **f** Overlay of the predictor gene expression in GSE38622 (training) and GSE223109 (test). **g** Prediction outcomes from tauFisher. **b**, **d**, **e**, **g** The dashed lines mark where predictions equal truth. RMSE: root mean square error. Source data are provided as a Source Data file.

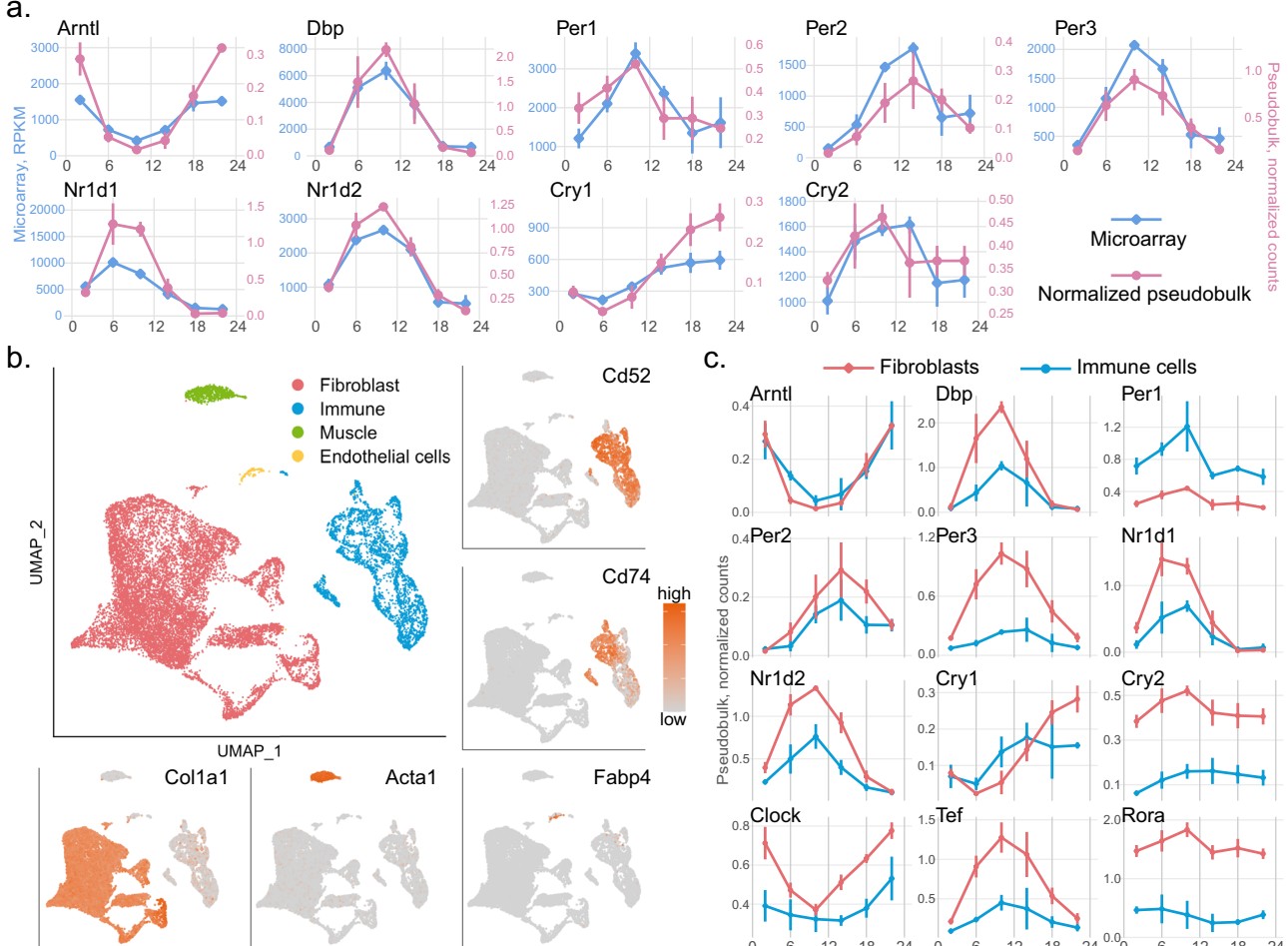

**Fig. 4 | The circadian clock is present in mouse dermal fibroblasts and immune cells. a** The normalized pseudobulk expression of the core clock genes generated from scRNAseq data (pink, *n* = 3 biologically independent samples per circadian time point) is consistent with their expression in the published microarray data (blue, *n* = 2 biologically independent samples per circadian time point, except that *n* = 3 at ZT2). Data are presented as mean values +/− SD. **b** Four major cell types, fibroblasts (red), immune cells (blue), muscle cells (green), and endothelial cells (yellow) were identified using canonical marker genes. Feature plots of

the representative marker genes are shown (orange: high expression; grey: low expression); *Col1a1* for fibroblasts, *Acta1* for muscle cells, *Fabp4* for endothelial cells, *Cd52* and *Cd74* for immune cells. **c** At the pseudobulk-level, expression pattern of the core clock genes is similar in fibroblasts (red) and immune cells (blue), while the amplitudes of the oscillations are dampened in immune cells for most of the core clock genes. n = 3 biologically independent samples per circadian time point. Data are presented as mean values +/− SD. Source data are provided as a Source Data file.

To validate the pseudobulk approach for studying the circadian clock in mouse dermis, we normalized the pseudobulk scRNAseq data and compared it with the published microarray data GSE38622 from mouse whole skin[34]. Overlay of the expression of nine core clock genes, *Arntl*, *Dbp*, *Per1*, *Per2*, *Per3*, *Nr1d1*, *Nr1d2*, *Cry1* and *Cry2*, reveals perfect consistency between the microarray data and the scRNAseq pseudobulk data (Fig. 4a), indicating that circadian clock gene expression in the dermis is captured in the pseudobulk data generated from scRNAseq data.

To study the circadian clock at a cell-type level in the skin, we integrated all samples and performed scRNAseq analysis to identify

cell types. In total, 16,866 cells passed the quality control, with around 950 cells per sample and around 2800 cells per ZT. Four major cell types, fibroblasts (*N* = 12,649), immune cells (*N* = 3353), muscle cells (*N* = 722) and endothelial cells (*N* = 142) were identified using canonical marker genes (Fig. 4b). Due to low cell counts for muscle and endothelial cells (*N* < 20) in some samples, we could not generate a reliable time series of pseudobulk data for these two cell types. Thus, we focus on the circadian clock in dermal fibroblasts and immune cells in this study.

In general, at the single cell level, the expression ranges of the core clock genes are similar in the two cell types, and the measurements of

the clock genes in fibroblasts are more variable (Supplementary Fig. 4). To compare the core clock in fibroblasts and immune cells, we computed and normalized the pseudobulk data for each of the two cell types in each sample. Both fibroblasts and immune cells possess robust circadian clock at the pseudobulk level. While the overall rhythms in the two cell types are consistent with each other, with core clock gene expressions peaking and troughing around the same time, the amplitudes of the oscillations are reduced in the immune cells compared to fibroblasts, indicating a dampened collective clock in immune cells (Fig. 4c). Whether this observation indicates less synchronous clocks in immune cells than in fibroblasts, or weaker clock function in each individual immune cell, is not known.

## Dermal fibroblasts and immune cells harbor different rhythmic pathways and processes

To study diurnal genes and pathways in dermal fibroblasts and immune cells, we used JTK_Cycle to identify rhythmic genes from the normalized pseudobulk data. We identified 1946 and 432 rhythmic genes in fibroblasts (Supplementary Data 1) and immune cells (Supplementary Data 2), respectively (Fig. 5a). The fewer rhythmic genes in immune cells is not caused by the lower cell count of immune cells, as randomly down-sampling the fibroblasts to the number of immune cells produced similar results. Only 79 genes were rhythmic in both cell types, with most of them related to the core clock network and metabolism.

Gene Ontology analysis revealed that rhythmic processes in fibroblasts and immune cells are different. Shared terms reflect basic cell integrity maintenance and function, including nucleocytoplasmic transport, regulations of cellular amide metabolic process, regulation of protein stability, and rhythmic process (Fig. 5b). For fibroblasts, additional metabolism processes and migration are significantly enriched in the rhythmic genes (Fig. 5b, red). For immune cells, the rhythmic genes enrich immune responses including defense response to virus, regulation of T-helper 2 cell differentiation, and response to interferon-beta (Fig. 5b, blue).

We selected some of the rhythmic genes in fibroblasts (Fig. 5c) and immune cells (Fig. 5d) and compared their expression patterns in the two cell types. For fibroblasts, we highlight genes related to glucose metabolism (*Pkm*), glycosylation (*Gal3st4*, *Plpp3*), oxidative phosphorylation (*Ndufs8*), collagen regulations (*Loxl2*, *Tgfb1*), amino acid metabolism (*Ivd*), sterol synthesis (*Scp2*, *Por*), and cell adhesion and migration (*Elmo2*, *Antxr1*), suggesting circadian regulation of the above processes at a molecular level (Fig. 5c, Supplementary Data 1). Interestingly, while some genes are only significantly rhythmic in fibroblasts because they are not expressed in immune cells (e.g. *Loxl2*), some are expressed at similar or higher levels in immune cells, but are not significantly rhythmic in the latter (e.g. *Ndufs8*, *Scp2*), indicating cell-type specific circadian regulations.

In the immune cells, genes related to inflammatory and immune response (*Cdk19*, *Cd84*), post-translational modification (*Sumo1*), extracellular matrix regulation (*Mmp9*), transcription regulation (*Med16*), electrochemical gradient maintenance (*Atp1b1*), and intercellular communication (*Stxbp6*) are rhythmic (Fig. 5d, Supplementary Data 2). We note that *Sumo1* is rhythmic in both fibroblasts and immune cells, but the expression peaks 4 hours later in immune cells than in fibroblasts.

Interestingly, the expression of *Il18r1* is significantly rhythmic with high amplitude in fibroblasts (*p*-value $= 2.21 \times 10^{-7}$), but not in immune cells (*p*-value $= 0.7104$) (Fig. 5c). The level of IL18, the ligand that binds to IL18R1, was found to be rhythmic in mouse peripheral blood[37]. Here, *Il18*, is significantly rhythmic in neither fibroblasts (*p*-value $= 0.3097$) nor immune cells (*p*-value $= 0.0925$) (Fig. 5d). But, it is possible that the insignificance of the *p*-value for immune cells is caused by noise introduced by summing the expression of all types of immune cells while it is mostly expressed in the myeloid cells.

To further explore the rhythmic pathways in dermal fibroblasts and immune cells, we divided the list of rhythmic genes into four groups based on their peaking time (Methods): day (ZT3 - ZT9), evening (ZT9 - ZT15), night (ZT15 - ZT21), and morning (ZT21 - ZT3 of the next day). The rhythmic genes are roughly evenly split: in fibroblasts, 426 peak during the day, 554 peak in the evening, 545 peak at night, and 421 peaks in the morning (Supplementary Data 1); in immune cells, 129 peaks during the day, 111 peaks in the evening, 87 peak at night, and 105 peaks in the morning (Supplementary Data 2). We then performed Gene Ontology analysis on the quarter-day rhythmic gene lists to identify the biological processes that are upregulated at different times of the day. We highlight some of the terms related to metabolism, signaling, cell proliferation and apoptosis, gene regulation, matrix regulation, and immune regulation (Fig. 5e).

During evening and night, when mice wake up, start feeding, and become active, processes such as the generation of precursor metabolites and energy, cellular respiration, and mitochondrial respiratory chain complex I assembly are upregulated in fibroblasts (Fig. 5e, Supplementary Data 3). Meanwhile, glycolytic processes are upregulated in fibroblasts, which is consistent with the finding that glycolysis is preferred at night in epidermal stem cells[38]. Additionally, similar to epidermal stem cells, more dermal fibroblasts may be in the S-phase of the cell cycle during the evening and night, as the DNA biosynthetic process is enriched during this time. Various signaling pathways are also enriched during this time, including the prostaglandin metabolic process and regulation of the apoptotic signaling pathway. Gene-regulatory mechanisms such as histone modification and mRNA splicing are upregulated during the evening and night in fibroblasts. Fibroblast migration peaks at night, which is consistent with previous findings that mouse wounds heal fastest during the active phase[39]. Immune regulation is also circadian regulated in fibroblasts, as terms including regulation of inflammatory response are enriched during this time. Compared to dermal fibroblasts during the evening and night, fewer terms related to metabolism, signaling, and gene regulation are enriched in dermal immune cells (Fig. 5e, Supplementary Data 4). But, almost all immune regulation terms such as defense response to the virus and interferon-beta production are upregulated in dermal immune cells during the evening and night, potentially contributing to shorter healing duration for wounds occurring during mice's active phase as well[39]. Additionally, such findings in mice imply that circadian regulation of immune response may be related to the more severe symptoms of inflammatory skin diseases, such as psoriasis, in the evening and at night[22,40].

In the morning and during the day, mice sleep and have lower food intake. Consistently, rhythmic genes peaking during this time in fibroblasts enrich for lipid catabolic process, glucose metabolic process, lipid storage, and response to starvation. Interestingly, extracellular matrix organization and cell-matrix adhesion peak during the day, possibly preparing for fibroblast migration, which peaks in the evening (Fig. 5e, Supplementary Data 3). For immune cells, rhythmic genes peaking during the morning and day generate fewer terms than the ones peaking during the evening and night, especially in the immune regulation category (Fig. 5e, Supplementary Data 4). Interestingly, rhythmic genes in the mouse dermal fibroblasts significantly enriched for genes linked to SNPs associated with systemic sclerosis[41], an inflammatory disease with increased collagen production by fibroblasts. Rhythmic genes in mouse dermal immune cells significantly enrich for SNPs associated with not only systemic sclerosis, but also vitiligo[42], a disease characterized by immune-mediated depigmentation of the skin (Fig. 5f).

In sum, we found that more genes are collectively rhythmic in fibroblasts than in immune cells, while only a few rhythmic genes are shared. Additionally, more metabolism processes are diurnally regulated in fibroblasts, with respiration peaking during the evening

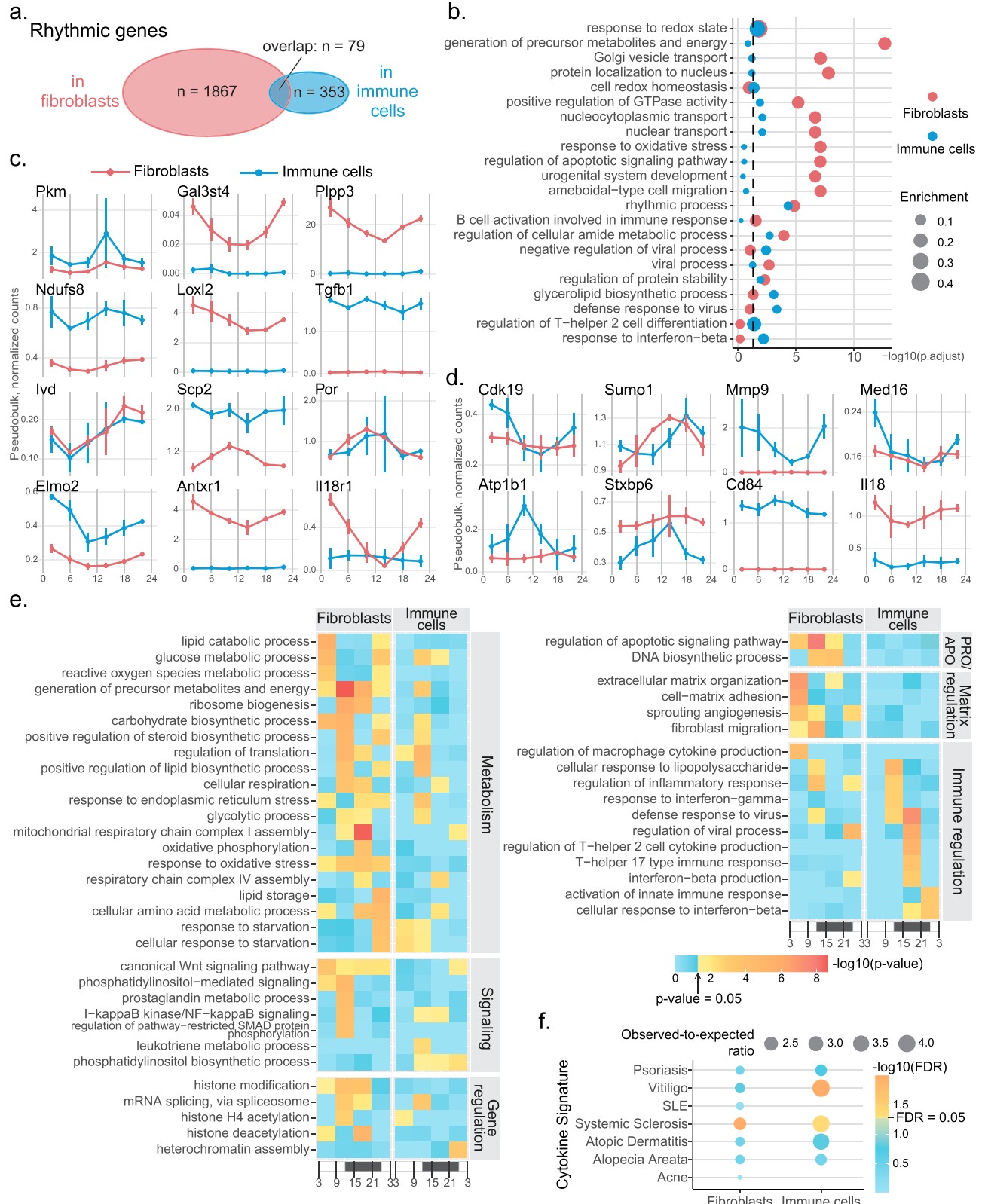

and night, and response to starvation and lipid storage peaking during the morning and day. On the other hand, immune regulation is almost exclusively upregulated by rhythmic genes that peak during the evening and night in immune cells. Importantly, rhythmic genes found in both fibroblasts and immune cells significantly enrich Genomewide Association Studies (GWAS) SNPs associated with human skin immune-mediated conditions, pointing to a potential

link between the skin circadian clock and autoimmune diseases of the skin.

### tauFisher determines that circadian phases are more heterogeneous in dermal immune cells than in fibroblasts

Analysis of the pseudobulk data from dermal fibroblasts and immune cells reveals dampened amplitudes of the core clock genes (Fig. 4c) in

**Fig. 5 | Different rhythmic processes are present in mouse dermal fibroblasts and immune cells. a** JTK_Cycle identified 1946 rhythmic genes in dermal fibroblasts (red) and 432 rhythmic genes in dermal immune cells (blue). Only 79 rhythmic genes are shared by the two cell types. **b** Gene Ontology analysis performed on rhythmic genes in fibroblasts (red) and immune cells (blue) reveals divergent biological processes being diurnally regulated in the two cell types. The dot size represents the enrichment score. The vertical dashed line marks adjusted *p*-value = 0.05. *P* values are determined using a hypergeometric test and adjusted using the Benjamini-Hochberg procedure. Expression of some of the rhythmic genes found in fibroblasts (**c**), and immune cells (**d**). *n* = 3 biologically independent samples per circadian time point. Data are presented as mean values +/− SD. **e** A heatmap showing *p*-values for some of the biological processes enriched by rhythmic genes peaking during each quarter-day time range. Color represents *p*-value. Blue: insignificant; yellow to red: significant with red representing a lower *p*-value. *P*-values are determined using hypergeometric test. x-axis represents time, with white being day and black being night. **f** Rhythmic genes in mouse dermal fibroblast and immune cells significantly enrich for genes within 200 kb of the GWAS signals of immune-mediated skin conditions. Blue: insignificant; yellow to orange: significant with orange representing lower FDR; FDR false discovery rate. Source data are provided as a Source Data file.

the immune cells and finds fewer rhythmic genes in immune cells than in fibroblasts (Fig. 5a). This observation could mean that each individual immune cell harbors weaker circadian clock, and/or the immune cells have more heterogeneous phases, so collectively they display a dampened clock. Note that variations of mean expression in single cells (vertical shifts of expression curves) do not cause dampened amplitudes at the pseudobulk level (Supplementary Fig. 5c, d), so this scenario is not considered in the following analysis.

To investigate the cause behind the dampened clock in dermal immune cells, we executed a bootstrapping approach that incorporates tauFisher for its ability to predict circadian time for transcriptomic data at different scales (Fig. 6a). Since the heterogeneity of a set of heterogeneous clocks should be captured at any given time point, we performed the analysis within each time point. The workflow involves the following steps: (1) trimming the scRNAseq data so that the expression matrix only includes the predictor genes identified in the training data and the cells labeled to be the interested cell types; (2) randomly sampling the same number of cells for each cell type to remove potential bias caused by different cell numbers, and summing the transcript counts in the pulled cells for each gene to create a pseudobulk dataset; (3) repeating the random sampling process (step 2) with replacement many times to create pseudobulk replicates for each cell type; and (4) predicting circadian time labels for the pseudobulk replicates using tauFisher. The idea is that if the cells harbor synchronous clocks, the pseudobulk replicates calculated from different rounds of sampling will be similar. In this case, the distribution of predicted time labels will be more concentrated. On the other hand, if the cells harbor heterogeneous clocks, the pseudobulk replicates calculated from the cells in different rounds of sampling will differ depending on which cells are pulled. The distribution of the prediction outcome in this case will be wider. Since the prediction outcomes are circular data, we then perform Rao's Tests for Homogeneity and Wallraff Test of Angular Distances to compare the mean and the dispersion around the mean.

To ensure that the pipeline works as expected, we generated simulated single-cell circadian gene expression datasets to represent a group of synchronized but dampened clocks (Supplementary Fig. 5a), and a group of out-of-phase but robust clocks (Supplementary Fig. 5b). As expected, the prediction outcome for the out-of-phase clocks has a significantly greater dispersion around the mean, indicating a more heterogeneous mixture of phases (Supplementary Fig. 5e, f).

We then applied the pipeline to the collected scRNA-seq data, focusing on the fibroblasts and immune cells. At each time point, we randomly selected *n* cells for each cell type, with *n* equal to 20% of the cell count of the cell type with the smaller population (immune cells in this dataset). Then, we used the exact same procedure to generate 500 pseudobulk replicates for the two cell types at each time point. tauFisher then predicts the circadian time for each pseudobulk replicate, yielding 500 predicted timestamps for each of the two cell types. We compared the distribution of the 500 predicted time labels of the two cell types at each time point.

In general, the prediction means are centered at different times for fibroblasts and immune cells (Fig. 6b), but around the predicted time for the whole-sample pseudobulk data (Fig. 3g). Whether one cell type's circadian clock is ahead of the other is inconclusive (Fig. 6c). Additionally, the distributions of the prediction outcome for immune cells are mostly multimodal and not as centered as the prediction distribution for fibroblasts (Fig. 6b). Indeed, the standard deviation of the prediction distribution is significantly greater for immune cells for five out of the six ZTs (Fig. 6c). This means that the bootstrapping pulled from a more heterogeneous population when sampling the immune cells, and thus implying that the clock phases are more heterogeneous in immune cells than in fibroblasts.

In sum, we were able to use tauFisher to obtain insights into the circadian heterogeneity for different cell types by predicting the circadian time for random samples from each of the cell types. We hypothesize that the circadian clock is more heterogeneous in dermal immune cells than in dermal fibroblasts, and such heterogeneity may be the reason behind the dampened core clock and fewer rhythmic genes we found in immune cells based on collective, cell-type level, gene expression data. Such a result is not unexpected, as the fibroblasts (Supplementary Fig. 6a, b) may be more homogeneous in their biological function than the immune cells, which contain dendritic cells as well as different types of macrophages and lymphocytes (Supplementary Fig. 6d, e) that serve different immune functions. Unfortunately, we did not capture enough cells for each specific fibroblast and immune cell type in the scRNAseq experiment to generate reliable pseudobulk data that is required for further circadian analysis (Supplementary Fig. 6c, f).

## Discussion

In this study, we developed tauFisher, a computational pipeline that accurately predicts circadian time from a single transcriptomic dataset and is applicable to within-platform and cross-platform training-testing scenarios. Particularly, tauFisher trained on bulk transcriptomic data accurately adds circadian timestamps for scRNAseq samples. This method allows investigators to place circadian timestamps on transcriptomic datasets and facilitates the determination of circadian time in the context of circadian medicine.

Most transcriptomic datasets in public genomics repositories lack circadian time labels, which complicates integration with or comparison to other datasets. Adding time labels for existing transcriptomic datasets is important, as the clock modulates the expression of many protein-coding genes; it is necessary to know whether a significant gene is truly differentially regulated by a condition or the expression appears to be different because the samples were collected at different times. Also, computationally adding circadian timestamps to existing transcriptomic datasets collected from various platforms, including from scRNAseq, opens up new possibilities for circadian research and allows investigators to take full advantage of the shared resource in an efficient and inexpensive way.

Circadian time determination is also a key step in the implementation of circadian medicine. To maximize effectiveness while minimizing side effects of treatments, it is necessary to take into consideration the patient's and relevant tissue's actual circadian time. For example, on-pump cardiac surgeries in the afternoon are less likely to cause perioperative myocardial injury than when conducted in the morning[43], and cancer radiation therapy in the morning causes less skin damage than in the afternoon[44].

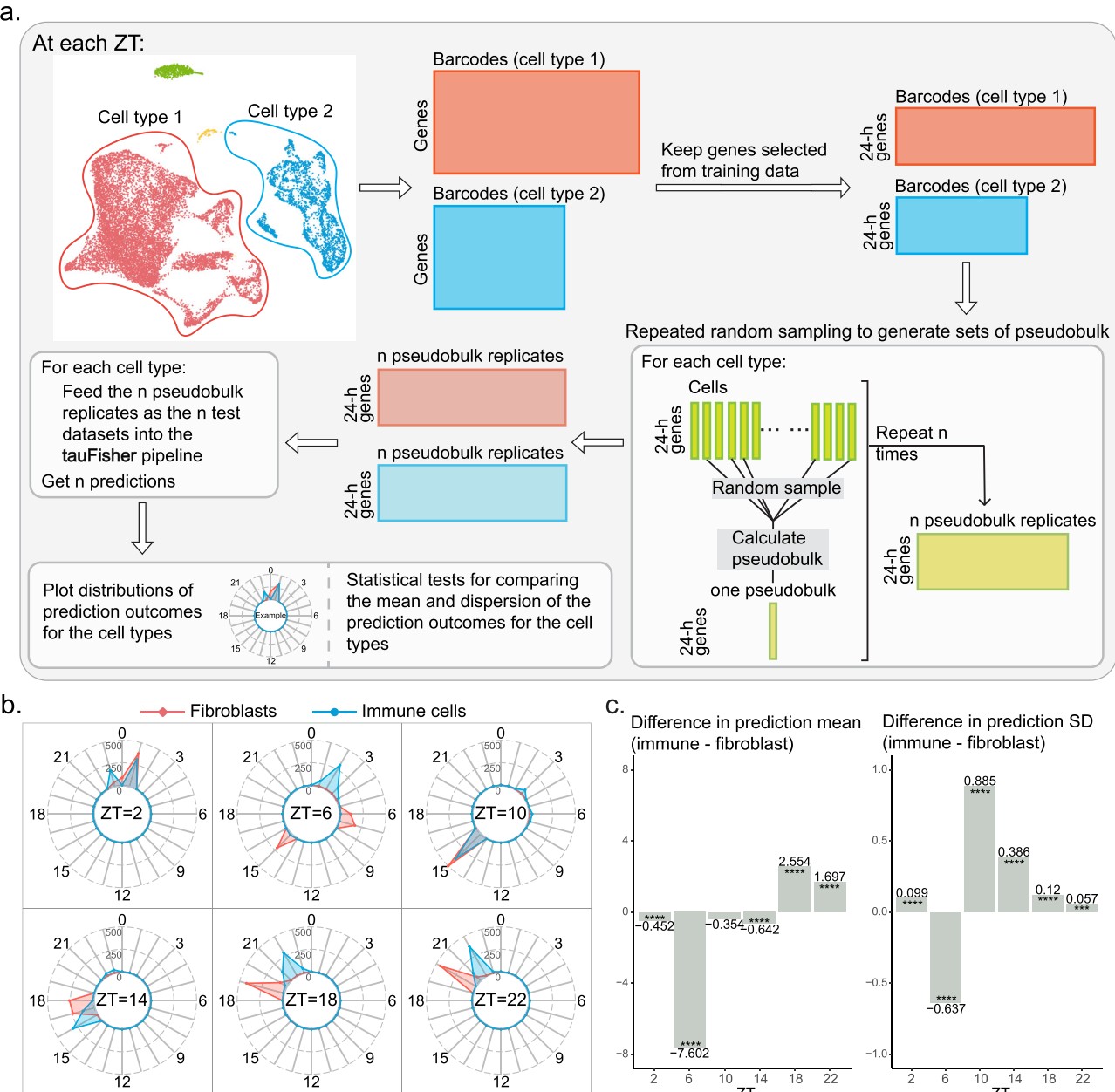

**Fig. 6 | tauFisher incorporated with bootstrapping suggests that the circadian phases in dermal immune cells are more heterogeneous than in dermal fibroblasts. a** A general overview of the pipeline that incorporates tauFisher with bootstrapping. **b** Radar plots showing the distribution of the prediction outcome for 500 pseudobulk replicates from dermal fibroblasts (red) and immune cells (blue). **c** Bar plots showing the differences between the prediction mean (left) and standard deviation (right) at each ZT (immune cells - fibroblasts). ****$p$-value ≤ 0.0001. *P*-values are determined using Rao's Tests for Homogeneity and the Wallraff Test of Angular Distances. Source data and exact *p*-values are provided as a Source Data file.

There have been several predictors of circadian time based on transcriptomic data[28–30], but to ensure the wide applicability of this approach, an assay platform-agnostic method requiring low number of test samples is desired. tauFisher is preferable as it can accurately determine circadian time from a single sample of transcriptomic data collected from various assay methods including scRNAseq.

Once trained, tauFisher requires a single transcriptomic sample to predict the circadian time. We examined tauFisher's ability to predict circadian time when the training and test data are from the same study, and we benchmarked it against state-of-the-art methods ZeitZeiger[28] and TimeSignatR[30]. tauFisher using one test sample achieved similar accuracy and RMSE for most of the datasets when compared to TimeSignatR which required two test samples that were 12 hours apart. ZeitZeiger failed to run for several of the datasets due to linear

dependency issues, and its success was dependent on the assay platform (Fig. 2, Supplementary Table 2). When it did run, ZeitZeiger achieved similar accuracy and RMSE as tauFisher, possibly because both predictors build on principal component analysis, suggesting that the molecular clock is well captured and represented by orthogonal linear combinations of predictor genes.

One of the most powerful features of tauFisher is its ability to accurately predict the circadian time when trained and tested on datasets collected from different assay platforms under different experimental settings. tauFisher achieves high accuracy and low RMSE in not only bulk-to-bulk cross-platform predictions but also bulk-to-scRNAseq predictions. While it is usually assumed that bulk RNAseq data are consistent with pseudobulk data calculated from scRNAseq data, it is necessary to verify this assumption in circadian studies. The

experimental settings for bulk and single-cell RNA extraction are different in terms of digestion duration and temperature, a factor that can alter the clock[4]. In this paper, we verify the consistency of circadian patterns in the two types of data by overlaying the expression of core clock genes (Fig. 4a); tauFisher's successful bulk-to-pseudobulk predictions (Fig. 3d, g) reassure such consistency. tauFisher outperformed ZeitZeiger and the two-sample TimeSignatR method, and the robustness in performance despite drastically different assay methods and experimental setups suggests that tauFisher captures and extracts the underlying biological correlations in gene expressions while minimizing the effects of the noise and variability introduced by subjects and technology.

Two key steps in tauFisher help achieve this: functional data analysis for training data and within-sample normalization for both training and test data. Functional data analysis for the training data enables tauFisher to remove minor noise, smooth the time expression curves, and generate the expression data between the sampled time points. The within-sample normalization step for both training and test data calculates the difference between each pair of predictor genes at a given time point so that the feature matrix is expanded while some baseline noise is removed. The differences between the genes are then re-scaled to be between 0 and 1 so that the data become unit-less. Doing so in parallel on the training and test datasets brings them individually to the same scale, instead of batch-correcting the train to the scale of the test or the opposite. This allows the testing of independent datasets without re-training. We note that this within-sample normalization is different from the within-subject normalization in TimeSignatR[30], which is based on mean expression calculated from multiple samples collected from more than one-time point over the circadian cycle.

Despite its unique circadian time prediction ability, tauFisher can be improved in several ways. While tauFisher performs well in terms of both accuracy and RMSE in almost all mouse benchmark datasets, its performance on human blood samples is subpar. Such performance can be attributed to greater human variability in expression patterns of the clock caused by hormones, stress, living style, and diet. A recently proposed pipeline called TimeMachine attempted to infer the circadian phase from a single human blood sample and reported 2-hour accuracy of 40 ~ 55%[45]. On the other hand, TimeSignatR using two human blood samples 12 hours apart achieved ~ 73% 2-hour accuracy (Fig. 2), emphasizing the necessity to address individual variability when predicting circadian phases for human samples and the effectiveness of the two-sample within-subject normalization step. One assumption of tauFisher's within-sample normalization step is that the differences between predictor genes are solely dependent on circadian time, and this step only removes uniform elevation or depression in expression values of clock genes. To further improve tauFisher's flexibility and performance on human data, it is worthwhile to incorporate a within-sample data centering step to address the differences caused by individual variability, while making sure that only one test sample is needed to make a prediction. Additionally, using tauFisher trained on healthy/control data to predict a time for data collected from a circadian-disturbed system only maps the disturbed test onto the timescale of the healthy samples. If one prefers to project test data onto the timescale of the diseased samples, a time series of transcriptomic data from diseased individuals is required. However, the circadian pattern is dampened in many diseases including cancer[46–51], making the expression data similar over time and thus more difficult to distinguish the time points. Although tauFisher showed promise in feeding-disturbed systems in skin and simulated dampened systems (Supplementary Figs. 2b and 5a, e), it would be valuable to further validate tauFisher on such datasets as they become available in future studies. Additionally, since uncertainty quantification may be particularly important when testing in disturbed systems, tauFisher can be improved to output confidence scores. Finally, while tauFisher accurately adds timestamps to unlabeled scRNAseq data and can predict

circadian time for pseudobulk data generated from a group of cells in scRNAseq data, it cannot overcome the high sequencing dropout rate in scRNAseq and thus cannot predict circadian time at a single cell level. Future work could focus on incorporating an imputation step into tauFisher to infer expression values of predictor genes. This step may not sacrifice computation and time efficiency greatly, as tauFisher only needs around 15 genes to make predictions.

In addition to testing tauFisher on published datasets, we also collected a time series of scRNAseq from mouse dermis. Consistent with previous findings[22,52], the circadian rhythm is robustly present in the dermis and the oscillatory patterns of the core clock genes agree with published data[34]. Comparing the rhythmic genes in fibroblasts and immune cells, we found that only a few rhythmic genes are shared by the two cell types and many pathways and processes are rhythmically regulated in a cell type-specific manner. Shared diurnally regulated terms include basic cellular functions and the rhythmic genes in fibroblasts have greater enrichment for metabolism-related terms, whereas the rhythmic genes in immune cells have greater enrichment for immune responses.

Combining tauFisher with other methods can guide the application of circadian medicine by providing additional insights and explanations of clinically observed circadian dysfunction. Dampened clock gene expression has been observed in psoriasis-affected skin[53,54], as well as in various types of cancer[46–51]. There is also evidence that restoring dampened circadian oscillations in diseased tissues can be effective in reducing cancer cell proliferation and tumor growth[49,51].

There are two possible behind-the-scene causes of dampened circadian rhythms at a bulk level: first, the circadian rhythm is dampened in every cell, but the cells are synchronous to each other (Supplementary Fig. 5a); second, the clock is normally functioning in every cell, but the cells are out of phase relative to each other (Supplementary Fig. 5b). Understanding which of the two scenarios is responsible for a dampened bulk-level clock gene expression is particularly important because in one case, it would be optimal to stimulate the clock to restore the circadian clock in the diseased tissue, while in the other case, synchronizing the clock is more suitable.

Here, we observed that the collective circadian rhythm in dermal immune cells is dampened compared to fibroblasts. We incorporated tauFisher with bootstrapping to investigate the cause behind the dampened collective circadian rhythm in dermal immune cells. tauFisher's prediction outcome suggests that the circadian phases in dermal immune cells are more heterogeneous than those in dermal fibroblasts, and this heterogeneity may contribute to the dampened rhythm in immune cells at a collective level. Due to technological constraints, our claim on differences in phase heterogeneity between dermal fibroblasts and immune cells relies on the computational analysis of the mouse skin scRNAseq and the simulated single-cell data.

The advantages tauFisher brings go beyond accurately adding timestamps when incorporated with other methods. For example, combining tauFisher with a batch-effect correction method may facilitate a cleaner integration and help minimize the effect of the circadian clock in transcriptomic data analysis. This approach harbors great potential as many efforts are going into integrating datasets from different studies to create metadatabases such as in the Human Cell Atlas.

In summary, tauFisher's consistent and robust performance in accurately predicting circadian time from a single transcriptomic data makes it a useful addition to the toolbox of circadian medicine and research.

## Methods
### tauFisher
tauFisher is a platform-agnostic method that predicts circadian time from a single transcriptomic sample. The method consists of three main steps: (1) identifying a subset of diurnal genes with a period

length of 24 hours, (2) calculating and scaling the difference in expression for each pair of predictor genes, and (3) linearly transforming the scaled differences using principal component analysis and fitting a multinomial logistic regression on the first two principal components.

**Preprocessing the expression matrix.** The first step in tauFisher is to average transcript measurements by genes such that the resulting matrix consists of unique genes. The subsequent training data consist of a gene expression matrix $X \in \mathbb{R}^{N \times P}$ with $N$ unique genes and $P$ samples, and a vector $\boldsymbol{\tau} \in \mathbb{R}^P$ of the corresponding time for each sample.

**Identification of periodic genes.** tauFisher specifies either the Lomb-Scargle[33] or JTK_Cycle[32] method in the `meta2d` function from the R package MetaCycle[55] to determine the periodic genes. It then selects the top ten statistically significant genes with a 24-hour period length. Using different numbers of diurnal genes does affect the prediction outcome. Although choosing the top ten does not guarantee the best performance, it is a safe and reasonable choice across different datasets (Supplementary Fig. 3c). These diurnal genes are then combined with the core clock genes that also have a period length of 24 hours to create the set of predictor genes $M$. The core clock genes for consideration in tauFisher are *Bmal1*, *Dbp*, *Nr1d1*, *Nr1d2*, *Per1*, *Per2*, *Per3*, *Cry1*, and *Cry2*.

**Data transfromation.** Subsetting the averaged expression matrix $X$ on the set of predictor genes $M$ yields averaged gene expression matrix $X' \in \mathbb{R}^{M \times P}$ with $M$ periodic genes and $P$ samples with known time ($\boldsymbol{\tau}$). The matrix $X'$ is then log-transformed element-wise: $X' = \log_2(X' + 1)$.

**Functional data analysis.** Since experiments have different sampling intervals throughout a circadian cycle, tauFisher uses functional data analysis to represent the discrete time points as continuous functions. This allows tauFisher to evaluate and predict the circadian time of the new samples at any time point and reduces the noise from the training samples.

Briefly, each gene $m$ has a log-transformed measurement at discrete time points $t_1, \ldots, t_P \in \boldsymbol{\tau}$ that may or may not be equally spaced. These discrete values are converted to a function $Z_m$ with values $Z_m(t)$ for any time $t$ using a Fourier basis expansion:

$$Z_m(t) \approx \sum_{k=1}^{K} c_{mk} \phi_k(t) \tag{1}$$

where $\phi_k(t)$ is the $k$-th basis function for $k = 1, \ldots, K$, and $\forall \, t \in \boldsymbol{\tau}$. $c_{mk}$ is the corresponding coefficient. The Fourier basis is defined by $\phi_0(t) = 1$, $\phi_{2r-1}(t) = \sin(r\omega t)$, and $\phi_{2r}(t) = \cos(r\omega t)$ with the parameter $\omega$ determining the period $2\pi/\omega$. Since the log-transformed data matrix $X'$ is non-negative, a positive constraint is imposed such that the positive smoothing function is defined as the exponential of an unconstrained function: $Y_m(t) = e^{Z_m(t)}$. The smoothing function also contains a roughness penalty to prevent overfitting. In practice, tauFisher sets the number of basis functions to $K = 5$, as it produces curves that are the most sinusoidal. Users can specify a different number of basis functions.

Although functional data analysis represents the discrete time points as continuous functions for each gene, tauFisher predicts circadian time at a user-defined time interval. By default, the time intervals are set to be one hour. The fitted functions $Y_m(t)$ are evaluated at the user-defined time interval to create the smoothed expression matrix $Y \in \mathbb{R}^{M \times T}$, where $T$ is the number of evaluated time points, and $\boldsymbol{\tau}_F \in \mathbb{R}^T$ is the new set of time points. If the time course of the samples spans less than 24 hours, then the fitted curves are evaluated hourly from [0, 23] such that $T = 24$ to ensure all 24 hours are evaluated. If the time course duration of the samples spans greater than 24 hours, then

fitted curves are evaluated from $[\min(\boldsymbol{\tau}), \max(\boldsymbol{\tau})]$ such that $T = \max(\boldsymbol{\tau}) - \min(\boldsymbol{\tau}) + 1$.

**Calculating and scaling the differences between each pair of genes.** For each time point, tauFisher generates all possible pairings of the selected predictor genes. It then calculates the differences between the two genes' functional data analysis-smoothed expression (stored in matrix $Y$). The resulting matrix retains differences between Gene A and Gene B (Gene A - Gene B) as well as between Gene B and Gene A (Gene B - Gene A). Then within each time point, the differences calculated from the gene pairs are scaled to be between 0 and 1 using the `rescale` function in R package scales. This way, 0 represents the minimum difference value and 1 represents the maximum. The formula is (value - min)/(max-min). As examples, we provided the resulting matrices for the training data (Supplementary Data 5) and the test data (Supplementary Data 6) when tauFisher was trained on GSE38622 and tested on GSE83855.

**The multinomial regression model.** The differences matrix is projected onto a lower dimensional space via principal component analysis, and the first two principal components become covariates $x_{i1}$ and $x_{i2}$ for observation $i$ in the multinomial regressor:

$$\log\left[\frac{P(\boldsymbol{\tau}_{Fi} = t | x_{i1}, x_{i2})}{P(\boldsymbol{\tau}_{Fi} = 0 | x_{i1}, x_{i2})}\right] = \beta_{t0} + \beta_{t1} x_{i1} + \beta_{t2} x_{i2} \tag{2}$$

All time points $t_1, \ldots, t_T \in \boldsymbol{\tau}_F$ are converted to be [0, 23], since time 0 is equal to time 24. Time zero, $\boldsymbol{\tau}_F = 0$, is set as the reference level in the model. The fitted multinomial regression model is then used to predict the circadian time of the new samples. We note that since time can be ordinal (accounting for the order) or continuous between [0, 24), we also tried other models such as an ordinal regression. However, these models were not as robust as the multinomial regression model and failed to run on the entire set of time points.

**Calculating prediction error**

To evaluate the performance of tauFisher, we need to calculate how close the predicted time is to the true time. Since the outcome is cyclic and ranges from [0, 23], we applied the following conversion to calculate the true difference $D$ from the difference $d$ between the predicted time and true time:

$$D = \begin{cases} d - 24, & \text{if } d > 12 \\ d + 24, & \text{if } d < -12 \\ d, & \text{if } -12 \leq d \leq 12 \end{cases} \tag{3}$$

**scRNAseq circadian gene expression simulations**

In Results, we demonstrated that tauFisher can be used to investigate circadian phase heterogeneity using simulated scRNAseq circadian gene expression data. We simulated three groups of data to represent three scenarios: (1) a group of synchronized but dampened clock genes, (2) a group of normal (robust) but asynchronous clock genes, and (3) a group of synchronized clock genes with normal amplitudes but variations in mean expression. For the three groups, the expressions of 9 representative diurnal genes over a time course of 24 hours are simulated using the following sine function:

$$y = A \sin\left(\frac{2\pi}{B}(x + C)\right) + D \tag{4}$$

where $A$ is the amplitude, $C$ is the phase shift, $D$ is the vertical shift, the period is $2\pi/B$, and $x$ is a sequence of integers from 0 to 23. We set $B$ to be 24, such that the period is $2\pi/24$, and $D$ to be a value big enough to ensure positive gene expression values. We used $D = 25$.

We used JTK_Cycle[32] to identify periodic genes, and its output contains inferred amplitudes and phase shifts for each gene. As inputs for our simulated datasets, we select the inferred amplitude and phase shift values for core clock genes *Bmal1*, *Dbp*, *Nr1d1*, *Nr1d2*, *Per1*, *Per2*, *Per3*, *Cry1*, and *Cry2* from Ref. 36. Then, for each dataset in Group 1, we simulate the expression of gene *i* as follows:

$$y_i = (A_i \times R_i) \sin\left(\frac{2\pi}{B}(x + C_i)\right) + D \quad (5)$$

where $R_i$ is one draw from a Beta(1, 2) distribution and all other parameters are as previously stated. Similarly, for each dataset in Group 2, we simulate the expression of gene *i* as follows:

$$y_i = A_i \sin\left(\frac{2\pi}{B}(x + C_i + R_i)\right) + D \quad (6)$$

where $R_i$ is one draw from a Normal(0, 6) distribution and all other parameters are as previously stated. For each dataset in Group 3, we simulate the expression of gene *i* as follows:

$$y_i = A_i \sin\left(\frac{2\pi}{B}(x + C_i)\right) + D + R_i \quad (7)$$

where $R_i$ is one draw from a Normal(0, 1) distribution and all other parameters are as previously stated. We generated 100 datasets for each group, which can be thought of as the simulated expression of 9 genes for 100 single cells over 24 hours. According to the simulations, only Group 1 and Group 2 scenarios can lead to dampened amplitudes at the pseudobulk level. Group 3 scenario is not considered as a possible cause of the dampened pseudobulk expression (Supplementary Fig. 5c, d).

We randomly select 6 time points without replacement over the course of the 24 hours to investigate circadian phase heterogeneity in the simulated data. At each time point, we randomly select 20% of the simulated single cells without replacement and sum their expression to obtain a pseudobulk dataset. We repeat this procedure 500 times for each of Group 1 and Group 2, generating 500 pseudobulk datasets per group. Then we used tauFisher to predict circadian time labels for the resulting pseudobulk datasets.

## scRNAseq experiments

**Mouse strains and husbandry.** The experiment is approved by the Institutional Animal Care and Use Committee (IACUC) at the University of California, Irvine under AUP-22-003. Wild-type male C57BL/6 mice were housed under a 12:12 light-dark cycle for two weeks prior to and during the time of the experiment. To collect telogen skin, mice were about 54 days old by the time of sample collection.

**Sample collection and sequencing.** Immediately after sacrificing a mouse with $CO_2$, hair on dorsal skin was removed with an electric razor and Nair Hair Removal cream. After the dorsal skin was isolated from the body, fat and remaining blood vessels were scrapped away with a scalpel blade. A circular piece of skin was obtained with a 12mm biopsy punch, and minced into tiny pieces. The minced skin was then digested with 2mL of a solution consisting of 0.27% Collagenase IV (Sigma, C5138), 10mM HEPES (Fisher Scientific, BP310-100), 1mM Sodium Pyruvate (Fisher Scientific, BP356-100), and 5U/mL DNase I (Thermo Scientific, EN0521) at 37 °C for 1.5 hours. The suspension was then filtered with 70 μm and 40 μm cell strainers to obtain single cells. SYTOX blue viability dye (1:1000; Invitrogen, S34857) was added to the cell suspension and live cells were sorted out using FACS at the UCI Stem Cell Research Center.

Samples were collected every four hours for three days to generate in total of 18 samples, providing three biological replicates per

circadian time point. The Chromium Single Cell 3′ v3 (10x Genomics) libraries were prepared and sequenced by the University of California Irvine Genomic High Throughput Facility with Illumina NovaSeq6000.

**Statistics and reproducibility.** No statistical method was used to predetermine the sample size. We chose to collect three biological replicates per circadian time point because previous circadian gene expression experiments showed that n = 3 allowed robust detection of circadian genes[6,34,35,56]. A random mouse was selected to sacrifice for each sample collection. No dataset was excluded from the analyses. During the collection and analysis of the scRNAseq data collected from mouse dermal skin, the investigators were not blinded to the time labels.

## Datasets and analysis

**Preprocessing for benchmark.** For GSE56931, we filtered out the time points related to the 38-hour continuous wakefulness and subsequent recovery sleep from the dataset provided in the TimeSignatR[30] package, to only include the 24-hour normal baseline time points. For GSE38622, the expression matrix was normalized as described[34]. For GSE157077, we used the transcriptomes of the mice who were fed normal chow through an entire circadian cycle (24 hours). We concatenated the three replicates of 24 hours to create one set of samples over 72 hours. For GSE54650, the raw CEL files for the kidney and liver were imported using the function `read.celfiles` in R package oligo. Each raw data matrix was then normalized with Robust Multiarray Average (RMA) using the function `rma`. To map the GPL6246 platform ID_REF to Ensembl transcript IDs, we used the transcript cluster ID and gene assignments listed in the table provided at https://www.ncbi.nlm.nih.gov/geo/query/acc.cgi?acc=GPL6246. For each transcript cluster ID, we removed all gene assignments unless they were Ensembl transcript IDs or started with Gm. If a transcript cluster ID was mapped to more than one gene, then we replicated that row by the number of genes (e.g., transcript reference ID 10344614 is assigned to three Ensembl transcript IDs so that row in the normalized dataset was replicated three times). The expression for each transcript cluster ID was then divided by the number of genes assigned (e.g., since transcript reference ID 10344614 has three gene assignments, the values for all three rows in the normalized dataset were divided by three). Transcript cluster IDs that were not assigned to any genes were removed from the normalized dataset. To convert the Ensembl transcript IDs to gene names, we use the R package biomaRt[57,58]. If biomaRt did not find a gene name, then we kept the original Ensembl transcript ID. For GSE54651, we converted the Ensembl gene IDs to gene names using the R package biomaRt[57,58]. If biomaRt did not find a gene name, then we kept the original Ensembl gene ID. For each time point in GSE117295 and GSE223109 (collected in this study), we summed the counts of each gene in all the cells without any pre-processing to create a pseudobulk dataset. In the case where the same gene occurs multiple times in the data, we took the mean of those entries. The resulting pseudobulk data at each time point is a single row vector in which each entry represents the expression value of a unique gene. The light-stimulated group is not considered in this paper.

**scRNAseq data analysis for dermal skin.** We used CellRanger version 3.1.0 with MM10 reference to process the raw sequencing output. The downstream analysis was done in Seurat V3 according to the vignette.

Cells with 850-7800 features and less than 13% of mitochondrial genes were kept. The SCTransform function was performed on each sample and 3250 integration features were selected using SelectIntegrationFeatures for each sample. Principal component analysis was then done on the integrated dataset and the Louvain algorithm was used to generate the clusters. Cluster identities were then determined in combination of marker genes found in the current clustering outcome and feature plots of canonical marker genes.

**Pseudobulk data analysis for dermal skin.** Pseudobulk data was calculated by summing the number of reads for each gene from all cells in a group. Normalized pseudobulk was calculated as transcript counts divided by total number of reads times 10,000. `meta2d` from the MetaCycle package was used on the pseudobulk data generated from the scRNAseq data collected from dermal skin to identify rhythmic genes. Genes with JTK_pvalue < 0.05 were determined to be significantly rhythmic. We used the meta2d_phase column to split the rhythmic genes into four groups based on their peaking time.

Gene Ontology analysis was performed using clusterProfiler in R with *p*-value < 0.05 as the significance cutoff.

### Enrichment for GWAS SNPs
For rhythmic genes with JTK_pvalue < 0.01 in the dermal fibroblasts or immune cells, we used the hypergeometric test to assess their enrichment among genes that are within 200kb of the GWAS signals of different skin immune-mediated conditions[41,42,59–63]. We used the transcripts expressed in the cells as the background gene list in the enrichment analysis. Significance cutoffs are false discovery rate≤0.05 and observed-to-expected ratio≥2.

### Statistics for circular data
The circular R package was used to perform statistical calculations and tests, including calculation of the mean and standard deviation, as well as the Rao's Tests for Homogeneity and the Wallraff Test of Angular Distances, for the circadian time prediction output in the Results section.

### Reporting summary
Further information on research design is available in the Nature Portfolio Reporting Summary linked to this article.

## Data availability
All published datasets used in this paper can be accessed through their respective GEO accession codes. The time series of microarray data collected from mouse kidney, liver, brainstem and cerebellum are available under GSE54650[56]. The time series of bulk RNAseq data collected from mouse kidney, liver, brainstem and cerebellum are available under GSE54651[56]. The time series of microarray data collected from mouse skin are available under GSE38622[34]. The time series of bulk RNAseq data from mouse SCN are available under GSE157077[36]. The time series of scRNAseq data from mouse SCN are available under GSE117295[35]. The bulk RNAseq data collected from mouse skin in control and time-restricted feeding conditions are available under GSE83855[6]. Although the datasets in[64] and[30] are both accessible through their GEO accession codes GSE56931 and GSE113883 respectively, this paper used the versions provided in the TimeSignatR package[30]: https://github.com/braunr/TimeSignatR. The time series of scRNAseq data from mouse dermal skin collected in this study are available in the GEO database under GSE223109. Source data are provided with this paper.

## Code availability
tauFisher is available as an R package at https://github.com/micnngo/tauFisher[65]. The two methods we compared tauFisher against are also available as R packages: TimeSignatR at https://github.com/braunr/TimeSignatR and ZeitZeiger at https://github.com/hugheylab/zeitzeiger.

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

## Acknowledgements

This project is supported by an award entitled "Skin Biology Resource-Based Center at UCI - Systems Biology Core" from the Center of Complex Biological Systems funded by NIH/NIAMS P30-AR075047 (J.D. and

M.N.); National Institute of Health grants R01AR056439 and P30AR075047 (B.A.); NSF grant DMS1763272 and a grant from the Simons Foundation (594598) (J.D., M.N., J.L. and B.A.); National Institute of General Medical Sciences, National Research Service Award GM136624 (J.D.); NSF grant DMS1936833 (M.N. and B.S.); the California Institute for Regenerative Medicine Training Program Award EDUC4-12822 (S.S.K.); and Chan Zuckerberg Initiative grant DAF2022-239946 (B.A., J.G., and L.C.T.).

## Author contributions

J.D., M.N., B.S., J.L. and B.A. conceived the project; J.D. and M.N. developed and implemented the tauFisher pipeline, and conducted the computational analysis; J.D. and S.S.K. collected the scRNAseq data. L.T. and J.G. contributed to the GWAS SNPs enrichment analysis. J.D. and M.N. wrote the manuscript. All authors edited and approved the manuscript.

## Competing interests

The authors declare no competing interests.
