## [Peer Review File · Nature Communications]

tauFisher predicts circadian time from a single sample of bulk and single-cell pseudobulk transcriptomic dataReviewer #1 (Remarks to the Author):

The manuscript by Duan et al. "tauFisher accurately predicts circadian time from a single sample of bulk and single-cell transcriptomic data" described a new computational pipeline called "tauFisher" to predict the circadian time of unlabeled single bulk and scRNAseq data. Overall, the manuscript is clearly written. TauFisher adds to a growing list of toolkits for the prediction of circadian time in transcriptome data. However, the unique aspects of tauFisher over existing methods were not clearly described. Moreover, I am less convinced of the validity of applying TauFisher in single-cell RNAseq data and the conclusion that dermal immune cells have heterogeneous phases as compared to fibroblast cells. The followings are my main concerns:

1. In the light of existing similar methods including CYCLOPS, ZeitZeiger, TimeSignatR, and so on, the authors should describe more clearly what exactly is unique in tauFisher. From the basic flowchart of tauFisher in Figure 1, it appears that tauFisher essentially utilizes PCA for the assignment of circadian time based on the normalized expression matrix of selected circadian genes (predictors), which is very similar to the procedure of CYCLOPS. So which features have been improved in TauFisher? The authors claimed that tauFisher is platform agnostic but did not show or explain the detailed evidences support this claim. Is this due to the unique normalization step used in the data preprocessing?

2. In TauFisher's application to scRNAseq data, only the analysis of pseudobulk data was used. The authors predicted the time of SCN scRNAseq data from a bulk time-series SCN transcriptome data. The accuracy of prediction is expected as the SCN scRNAseq was treated as a bulk or pseudo-bulk. This basically reflects the consistency between time-series bulk SCN RNAseq data and time-series SCN scRNAseq data when considered as a whole. The authors made further attempts to estimate the phase heterogeneity of dermal immune cells and fibroblast cells from time-series skin scRNAseq data. However, the means and standard deviations were calculated from the 500 randomly sampled pseudo-bulk replicates (Figure 6b-c). This does not necessarily reflect the phase diversity in the real single-cell data. The fact that the mean phase of immune cells is advanced compared to fibroblast at some ZT times but delayed at other ZT times (Figure 6b) indicates that the fluctuation of phases is more due to the estimation procedure itself rather than the biological differences. As mentioned in the paper, the circadian clock genes often have drop-outs and their expression values are noisy in scRNAseq data. While it is easy to estimate the phases from the bulk data, it would be hard to estimate the real phases of single cells without considering the intrinsic noise due to low copy number of mRNA in individual cells, the limited detection in sequencing-based transcriptomic analysis, and further internal cellular heterogeneity within a cell type. Therefore, tauFisher estimation of phase diversity within scRNAseq data appears over-simplified. The claim that dermal immune cells have more heterogeneous phases than fibroblast cells requires experimental justification.

Reviewer #2 (Remarks to the Author):

Duan et al present a new algorithm, tauFisher, a "platform agnostic" circadian prediction algorithm. Given well annotated training data from a tissue of interest it aims to infer circadian phase in a new sample from the same tissue/cell type - even if obtained in a different platform or individual. They then apply this method to some newly collected single cell data and attempt to assess phase synchronization.

This (and related) problems are of increasing interest. I think this work is a significant contribution - I complement the authors.

Nonetheless I have some concerns. Much of this likely can be accommodated by the authors simply better describing the limits and assumptions of their method. But I also have some more substantive concerns about the benchmarking - and their final use case. All in all - I think this work has value - and I suspect by concerns can be relatively easily addressed.

General Method:

Much of the improved robustness claimed by TauFisher appears to be attributed to its within sample normalization.

This seems like an interesting approach.

But I think there are several assumptions underlying this normalization. They should probably should be more explicitly articulated/discussed

(1) In taking the differences between genes(i) and gene(j) in every sample - I think you are effectively assuming that those differences only depend on circadian time (and random noise) If there are systematic difference between the average levels of these genes between subjects (as might be expected in human data) - or between batches - this normalization would not address this? Similarly wide variation in amplitude among genes in people/subjects

(or course less concern with mouse data)

(2) Similarly, if you try to apply your model to a perturbation that may affect the amplitude or baseline expression of your genes (as many perturbations do) your training data on non-perturbed systems might not be appropriate?

(3) In assuming that the rank differences between genes i,j are roughly comparable between platforms - you are assuming that all measures all roughly scale the same amount - no? If the scaling factor for Per2 from Array -> Seq is X 6, Arntl is x2 and CRY1 is X 10 - the rank ordering of differences will be change.

#####

Benchmarking

(1) In terms of the mouse tissue benchmarking Liver and Kidney (and then Lung) had the most robust rhythms in Zhang et al

(at least in terms of numbers of cycling genes - and my guess is in amplitude)

SCN is clearly one of the most robust brain tissues.

Could you evaluate tau fisher on the non-scn brain tissues in the Zhang data...

Help readers see how it does on "weaker tissues."

(2) In terms of the blood data - I could not tell if the individual subjects contributed to both the training and testing samples in each run

Or if you made sure to train and test on disjoint subjects (rather than disjoint samples). It should be done that way. If it was already - great. If not pleas redo

(3) Also in terms of blood data - this seems like a good venue to test your cross-platform abilities Can you provide data showing what happens when you use the model trained on Arnardottier et al on the Braun data (and vice versa)

I think this would help assess tauFisher in some more realistic use cases - and easy to do given your work already!

I don't think you ever define "Accuracy"

I gather you are using as tauFischer is a classifier and provides discrete labels rather than continuous times - and is just percent assigned to correct label?

But I couldn't find in manuscript.

For the data that had 2,3, or 4 hour resolution, was the TauFischer multivariate logistic model limited to those possibilities? Or did you keep 24-hour resolution - and only count correct if it got the exact right time?

On blood TauFischer (and the other methods) seem to have a <50% accuracy, and an RMSE error near 5-6 hours (+/- 5 hour window is a pretty big range)

You mention the puzzling performance attributed to TimeSignatR - it makes me wonder if it was used correctly?
As you note in you note in the discussion the "2 point" version did worse than the 1 point?
Could you explain what you mean when you say the original paper used the test data in training?
I am not trying to cause a feud! So if you prefer to leave out of paper that is fine!
But it would help me feel more confident that assessment was done fairly

I don't know the red stars are on the plots

You mention a new method Tempo that estimates prediction uncertainty
Is this easy to benchmark? - this is not required - you have already tested 2 other methods which I take as a good faith effort.
But if you can't - I wonder - there is a wide variety in the prediction accuracy that you do show.
Can you guide the user as to how to estimate the fit confidence of Tau Fisher in a new application

#####

New Application

Most of the analysis of the new scRNA-seq dermal skin data was of course done without tauFisher - as collection time was known.

The interesting analysis that used tau-fischer was based on using TauFisher to try to assess if the reduced amplitude in immune cells as opposed to fibroblasts was due to desynchrony vs a lower amplitude in each cell

The authors take subsamples of the single cell pseudobulk - and assess the variability in their phase estimates /do a bootstrap analysis.

They find that the estimates are more spread out in the immune cells and relate this to decreased synchronization.

(1) It seems to me that if you assumed significant random variation in the DC offset term (the D component of your sinusoidal variation model for each gene) this would also give you more variability in your phase estimate. Why do you assume variability in phase among cells is a better explanation than simply more variation in mean expression among these cells?

(2) A lesser point - my understanding is that the measurement of more lowly expressed transcripts are also innately more variable in scRNAseq - so wouldn't you expected to see more variable measurements anyway?

#####

For the enrichment analysis - what was the background list used? Transcripts expressed in the cell - or the whole genome?

#####

Discussion

Some of the points you rightly emphasize are in eventual application for circadian medicine.

Like learning cardiac time given myocardial injury data

Similarly adding "time" information to public data repositories

But I think you should probably emphasize that tauFisher would need training data for that human tissue - and likely training data unique to any disease state.

Also that the +/- 5 hour window they give (RMSE) for the human blood data - might suggest that human variability remains significant obstacle

I know the authors know this - but given the discussion points it should be explicit.

Also some more focused attention to where you think the weakness of tau fischer are warranted.

#####

Title:

A fairer title might be

"tauFisher accurately predicts circadian time from a single sample of bulk or pseudobulk transcriptomic data"

You never really do assign phase to a single-cell ..

Finally, I must admit I have not tested code.

It seems to all be there- and it looks pretty - but I have not run it myself.

I asked a student to run it (and he was able to)

Reviewer #3 (Remarks to the Author):

TauFisher is presented as an algorithm to predict circadian time from a single experiment transcriptomes, including from single-cells. The method has a number of interesting features: (1) it is applicable independent of the transcriptome platform and apparently performs well in a cross platform setting; (2) does not require the training data to be a complete time series; (3) it uses comparisons of clock genes within a sample; (4) it can work for single cell data. Overall the presentation of some of the analyses needs to be improved and in general the methods need to be better explained.

Results 2.1

1. The method part with the difference matrix is not well explained, particularly the scaling. This matrix should be shown and compared across datasets. How does it look in broken clock conditions?

Results 2.2

2. The presentation of Figure 2 is very busy, I would suggest removing all the p-values. The red stars are not described, what are those?

3. It seems that TauFisher performs worse than Zeitzeiger in many cases. Was this expected?

4. TauFisher seems to be sensitive to the seed genes. How is the performance dependent on the number of seed genes?

5. The accuracy in blood is always quite low, can the authors comment on it?

6. Which were the selected genes in the different cases? Are the genes the same for the three methods? What if the same genes are used in all three methods?

7. It would be useful to show RMSE and Accuracy on training and testing sets separately.

Results 2.3

8. Fig3A. Axis labels are missing, what are the units? It should be shown in log2 scale as the fits were done in log-scale. In general, the authors should make sure that there is no mix-up of linear and log scale.

9. A scatter plot of predicted vs true sample time should be plotted

10. The right circular plots are difficult to interpret and not informative

Results 2.4

11. These sentences are very confusing: "The test data appears to be noisier since it is not normalized. tauFisher does not require the test data to be normalized as the within-sample normalization step is part of the pipeline. Explain better.

12. A scatter plot would be more informative and easier to read than Table 2.

13. It is not so surprising that pseudo-bulk patterns are similar to bulk, this is most typically the case in scRNA-seq

Results 2.5/2.6

14. While these sections are potentially interesting, these analyses seem unrelated with the rest of the paper (tauFisher). It seems like these sections should go in a separate manuscript, or they will not be noticed inside a methods paper.

Results 2.7

15. The bootstrapping procedure is quite complex and should be better described.

16. The authors should clarify the argument behind "indicating that immune cells have a more heterogeneous clock phase than fibroblasts."

REVIEWER COMMENTS

Reviewer #1 (Remarks to the Author):

The manuscript by Duan et al. "tauFisher accurately predicts circadian time from a single sample of bulk and single-cell transcriptomic data" described a new computational pipeline called "tauFisher" to predict the circadian time of unlabeled single bulk and scRNAseq data. Overall, the manuscript is clearly written. TauFisher adds to a growing list of toolkits for the prediction of circadian time in transcriptome data. However, the unique aspects of tauFisher over existing methods were not clearly described. Moreover, I am less convinced of the validity of applying TauFisher in single-cell RNAseq data and the conclusion that dermal immune cells have heterogeneous phases as compared to fibroblast cells. The followings are my main concerns:

We appreciate the reviewer for the insightful comments and suggestions. We have followed the suggestions and edited the manuscript accordingly.

1. In the light of existing similar methods including CYCLOPS, ZeitZeiger, TimeSignatR, and so on, the authors should describe more clearly what exactly is unique in tauFisher.

Among these three methods, CYCLOPS outputs the relative ordering of unordered transcriptomic datasets, whereas ZeitZeiger and TimeSignatR output circadian timestamps for the test data, making these two methods more similar to tauFisher. However, existing methods run into linear dependency issues, require at least two test data a few hours apart to make good predictions, or require retraining for every prediction. We now describe some of the limitations of the previous methods more clearly in the Introduction:

"... But, they have limitations. CYCLOPS outputs the relative ordering instead of timestamps of samples and requires reconstruction to incorporate every new sample. ZeitZeiger frequently runs into linear dependency issues, needs to be retrained before each prediction, and is not generalizable between transcriptomic platforms. BIO_CLOCK does not require re-training for each prediction but is not time-efficient. TimeSignatR performs well if there are two test samples, but rough knowledge of the test samples' time labels is required, and its performance depends on the time interval between the two samples."

For within-platform predictions, the most unique feature of tauFisher is that it only needs to be trained once on a time series of transcriptomic data to accurately predict a circadian timestamp for a single external transcriptomic data from the same tissue.

For cross-platform performance, tauFisher outperforms ZeitZeiger and timeSignatR as we have demonstrated in the manuscript (Figure 3, Results). Unprecedentedly, tauFisher trained on bulk RNAseq or microarray data can accurately predict circadian timestamps for scRNAseq data, making tauFisher an effective framework for expanding existing scRNAseq database for circadian research and for facilitating integration and comparison of scRNAseq datasets. We demonstrated tauFisher's high accuracy for adding time labels to scRNAseq data in both the SCN (the central clock, Figure 3) and the skin (a peripheral clock, Figure 3).

We also summarized tauFisher's key features in the Introduction:

“tauFisher improves on previous methods in several ways: (1) it does not require the training data to be a complete time series; (2) the within-sample normalization step allows tauFisher to give an accurate prediction from just one sample; (3) since tauFisher only needs a few features to make accurate predictions, training and testing are computationally efficient; (4) tauFisher is platform agnostic and users only need to train the predictor once and can use the same predictor to make predictions for external datasets of the same tissue, regardless of the platform; and (5) unprecedentedly, tauFisher trained on bulk sequencing data is able to accurately predict the circadian time of single-cell RNA sequencing (scRNA-seq) data, and it can be used to investigate circadian phase heterogeneity in different cell types.”

From the basic flowchart of tauFisher in Figure 1, it appears that tauFisher essentially utilizes PCA for the assignment of circadian time based on the normalized expression matrix of selected circadian genes (predictors), which is very similar to the procedure of CYCLOPS.

CYCLOPS and tauFisher both use a dimensionality-reduction technique to extract features in the data, and in that sense, we agree with the reviewer that the two methods are similar. However, we argue that CYCLOPS and tauFisher are very different pipelines, since at the core, CYCLOPS uses a combination of PCA and autoencoder neural network (Anafi et al., 2017), whereas tauFisher uses PCA and multinomial regression. Additionally, CYCLOPS is designed to output a relative ordering of unlabeled samples (Anafi et al., 2017) whereas tauFisher is designed to output timestamps for test sets.

CYCLOPS (Anafi et al., 2017) first scales and normalizes the data, then uses a singular value decomposition approach to reduce the dimensionality of the data. It keeps enough eigengenes to ensure that 85% of the variance of the data is preserved, and it inputs this decomposition into the autoencoder, which is designed specifically for temporal reconstruction to capture the circadian patterns in the data. CYCLOPS specifically uses the identity function between the layers (nonlinear) and in the bottleneck layer and a special circular neuron activation scheme to handle circular data. CYCLOPS, then, combines a linear encoding/decoding with a coupled circular bottleneck node. Additionally, CYCLOPS requires the input data from the entire periodic cycle to form an ellipse and its orderings are relative. While CYCLOPS assigns phases, additional information is needed to assign a circadian time such as acrophase.

tauFisher uses PCA, a completely linear method, to reduce the dimensionality of the data. Furthermore, because tauFisher does not require complete time series data or a shape, it is more flexible.

We updated the Introduction to provide more details about CYCLOPS:

“Several groups have developed methods to infer circadian time of a sample (organism, organ, or tissue) based on transcriptomic data. CYCLOPS [26, 27] uses an autoencoder neural network to infer circadian phases by ordering the data collected from the entire periodic cycle. ZeitZeiger...”

So which features have been improved in TauFisher?

As mentioned above, we provided an overview of tauFisher’s key features in the Introduction:

“tauFisher improves on previous methods in several ways: (1) it does not require the training data to be a complete time series; (2) the within-sample normalization step

allows tauFisher to give an accurate prediction from just one sample; (3) since tauFisher only needs a few features to make accurate predictions, training and testing are computationally efficient; (4) tauFisher is platform agnostic and users only need to train the predictor once and can use the same predictor to make predictions for external datasets of the same tissue, regardless of the platform; and (5) unprecedentedly, tauFisher trained on bulk sequencing data is able to accurately predict the circadian time of single-cell RNA sequencing (scRNAseq) pseudobulk data, and it can be used to investigate circadian phase heterogeneity in different cell types.”

The authors claimed that tauFisher is platform agnostic but did not show or explain the detailed evidences support this claim.

We define a prediction pipeline as platform agnostic if it works well when trained and tested within different sequencing platforms, and also gives accurate predictions when the training and test data are from different platforms.

In the manuscript, we benchmarked tauFisher against ZeitZeiger and TimeSignatR to demonstrate its better performance when trained and tested within bulk RNAseq or microarray data in multiple tissue types (Figure 2, Results). For cross-platform bulk-to-bulk predictions, we showed tauFisher’s high accuracy when trained on mouse skin microarray data and tested on mouse skin bulk RNAseq (Figure 3, Results). For cross-platform bulk-to-scRNAseq pseudobulk predictions, tauFisher trained on bulk RNAseq or microarray data can give accurate timestamps for scRNAseq samples from mouse SCN and mouse dermal skin respectively (Figure 3, Results). Based on these results, we determined that tauFisher is platform agnostic.

Is this due to the unique normalization step used in the data preprocessing?

We agree with the reviewer that the within-sample normalization step in the data processing step allows tauFisher to perform well for multiple assay platforms and cross-platform. The first step of the within-sample normalization is to calculate the differences between the expression level of predictor genes within each sample. This step partially removes universal elevation/depression in the measurement due to biological or technical differences, and captures the relationships between the predictor genes. The second step is to rescale the differences to be between 0 and 1. This step converts the expression data to a ranking data, and thus provides a unitless representation of the relationships between the predictor genes in each sample, bringing all data to the same scale.

In the Discussion, we discussed the advantages of using this within-sample normalization step:

“The within-sample normalization step for both training and test data calculates the difference between each pair of predictor genes at a given time point so that the feature matrix is expanded while some baseline noise is removed. The differences between the genes are then re-scaled to be between 0 and 1 so that the data become unit-less. Doing so in parallel on the training and test datasets brings them individually to the same scale, instead of batch-correcting the train to the scale of the test or the opposite. This allows testing of independent datasets without re-training.”

2. In TauFisher's application to scRNAseq data, only the analysis of pseudobulk data was used. The authors predicted the time of SCN scRNAseq data from a bulk time-series SCN transcriptome data. The accuracy of prediction is expected as the SCN scRNAseq was treated

as a bulk or pseudo-bulk. This basically reflects the consistency between time-series bulk SCN RNAseq data and time-series SCN scRNAseq data when considered as a whole.

We agree with the reviewer that tauFisher does not predict circadian time for each individual cell in a scRNAseq dataset. To avoid confusion, we have changed the title of the manuscript to

“tauFisher accurately predicts circadian time from a single sample of bulk or single cell pseudobulk transcriptomic data”.

We trained tauFisher on bulk SCN RNAseq and tested on pseudobulk data calculated from SCN scRNAseq data to (a) verify the assumption that the pseudobulk data is consistent with the bulk RNAseq data and (b) test whether tauFisher can add time labels to scRNAseq samples when trained on bulk transcriptomics data.

We needed to verify the consistency between bulk RNAseq data and scRNAseq pseudobulk data because the experimental protocols to obtain these two types of data are different, and such differences could be important in circadian studies. In bulk RNAseq experiments, the tissue samples are usually snap-frozen in liquid nitrogen as soon as they are excised from the body. The samples are then ground and lysed for RNA extraction. The whole process usually takes around 30 minutes. One can also skip the snap-freezing step and obtain bulk RNAseq from fresh tissue. In contrast, in scRNAseq experiments, the tissue samples go through multiple incubations in different digestion solutions at 37C until live cells are dissociated from each other to form a single cell suspension. The suspension is then usually kept on ice for dead cell removal. The whole process takes a few hours. As temperature can affect the clock (Buhr et al., 2010) and the preprocessing of the samples before sequencing takes different amount of time, it is necessary to verify that a time series of bulk RNAseq and a time series of scRNAseq pseudobulk data are consistent with each other. One way to verify this is to overlay the bulk transcriptomics data with the pseudobulk data. For the SCN data, such consistency is shown in Wen et al., 2020. For the skin, we demonstrate such consistency in Figure 4a in this manuscript. The fact that tauFisher trained on bulk RNAseq/microarray data can accurately predict timestamps for scRNA-seq pseudobulk data reassures such consistency. These observations also suggest that clock gene expression represents the clock at the time of sample isolation in both bulk RNA and scRNA extractions.

To highlight the importance of these considerations, we added the following text to the Discussion:

“While it is usually assumed that bulk RNAseq data are consistent with pseudobulk data calculated from scRNAseq data, it is necessary to verify this assumption in circadian studies. The experimental settings for bulk and single cell RNA extraction are different in terms of digestion duration and temperature, a factor that can alter the clock [4]. In this paper, we verify the consistency of circadian patterns in the two types of data by overlaying the expression of core clock genes (Figure 4a); tauFisher’s successful bulk-to-pseudobulk predictions (Figure 3) reassure such consistency. However, tauFisher is the only method that successfully predicted the circadian time for the pseudobulk data (Figure 3) despite the consistency between bulk RNAseq/microarray data and scRNAseq pseudobulk data. The robustness in performance despite drastically different assay methods and experimental setups suggests that tauFisher captures and extracts the underlying biological correlations in gene expressions while minimizing the effects of the noise and variability introduced by subjects and technology.”

The authors made further attempts to estimate the phase heterogeneity of dermal immune cells and fibroblast cells from time-series skin scRNAseq data. However, the means and standard deviations were calculated from the 500 randomly sampled pseudo-bulk replicates (Figure 6b-c). This does not necessarily reflect the phase diversity in the real single-cell data. The fact that the mean phase of immune cells is advanced compared to fibroblast at some ZT times but delayed at other ZT times (Figure 6b) indicates that the fluctuation of phases is more due to the estimation procedure itself rather than the biological differences.

We want to clarify that for each cell type, the means and standard deviations were calculated from the 500 timestamps predicted by tauFisher from 500 randomly sampled pseudobulk replicates, instead of directly from the pseudobulk replicates themselves.

The estimation procedure was exactly the same for the fibroblasts and immune cells, so it is unlikely that the estimation procedure caused the difference we observe in the two cell types. To eliminate the possible bias caused by cell number, for each round of random sampling at each time point, we pulled out the same number of cells for both the fibroblasts and the immune cells. The number of cells to pull was determined to be 20% of the cell type with the lower cell count. For the case in the manuscript, the number of cells to pull in each round of bootstrapping for each cell type is 20% of the cell count of immune cells. Then the pseudobulk data was calculated for each group of randomly pulled cells and circadian timestamps were predicted by tauFisher. The exact same process was repeated 500 times with the same parameters for the two cell types at each time point. The differences between the two cell types reflect biological differences in clock features between the two cell types, instead of the estimation procedure.

We edited the text to clarify the two points above:

“We then perform the pipeline on the collected scRNA-seq data, focusing on the fibroblasts and immune cells. At each time point, we randomly selected n cells for each cell type, with n equal to 20% of the cell count of the cell type with the smaller population (immune cells in this dataset). Then, we used the exact same procedure to generate 500 pseudobulk replicates for the two cell types at each time point. tauFisher then predicts the circadian time for each pseudobulk replicate, yielding 500 predicted timestamps for each of the two cell types. We then compared the distribution of the 500 predicted time labels of the two cell types at each time point.”

We tested this procedure on simulated data (Supplementary Figure 5) to validate our idea of incorporating tauFisher with bootstrapping to gain insights into circadian phase heterogeneity. We simulated 100 cells with variations in amplitude and 100 cells with variations in phase. Both of the groups display similar dampened amplitude when observed at a pseudobulk level. Using the procedure described above and in the manuscript, we obtained 500 predictions for each group of the cells. Comparing the means and standard deviations of the prediction outcome, we see that as expected, the group with variations in phase have significantly higher standard deviation in the circadian time prediction outcome. Additionally, similar to what we observed in the real data, the mean for the group with variations in phase is advanced or delayed at different ZTs when compared to the group with variations in amplitude, while we made sure that the cells from these two scenarios display very similar circadian patterns at the pseudobulk level (Supplementary Figure 5).

As the same process was done on the two cell types, the heterogeneity estimation process worked as expected on the simulated data, and we observe similar patterns in the real and the

simulated data, we argue that the differences in the standard deviation in the prediction outcomes are not due to the estimation procedure.

As mentioned in the paper, the circadian clock genes often have drop-outs and their expression values are noisy in scRNAseq data. While it is easy to estimate the phases from the bulk data, it would be hard to estimate the real phases of single cells without considering the intrinsic noise due to low copy number of mRNA in individual cells, the limited detection in sequencing-based transcriptomic analysis, and further internal cellular heterogeneity within a cell type. Therefore, tauFisher estimation of phase diversity within scRNAseq data appears over-simplified.

We agree with the reviewer that the noise and high frequency of sequencing drop-outs in scRNAseq data make it difficult to predict circadian phase at a single cell level, and we do not claim that tauFisher can make such predictions. Instead, for all of the predictions on scRNAseq data, we took a pseudobulk approach to overcome such difficulties due to the sequencing technology. We discussed this in the Results:

“Due to the frequency of sequencing dropouts for clock genes in scRNA-seq data, investigating the circadian clock within each cell is not yet achievable. To overcome this limitation, previous studies have used pseudobulk approaches to investigate the clock in scRNAseq data.”

As previously mentioned, to avoid confusion, we have also changed the title of this manuscript to:

“tauFisher accurately predicts circadian time from a single sample of bulk or single cell pseudobulk transcriptomic data”.

To compare phase heterogeneity between cell types, we used the pseudobulk approach applied to individual cell types instead of to the entire sample. The idea is that, if we pull from a group of synchronous clocks (for example, all cells are at ZT10), then the pseudobulk calculated from the pulled cells will be very similar in the 500 rounds, and thus the prediction outcome will have a very narrow distribution. On the other hand, if we pull from a group of heterogeneous clocks (for example, 20% at ZT10, 20% at ZT12, 50% at ZT23 and 10% at ZT6), the pseudobulk data calculated from the cells in each round of pulling will differ depending on which cells are pulled. The distribution of the prediction outcome in this case will be wide.

To ensure that noise in data is not causing bias in the final conclusion, we randomly sampled cells to calculate pseudobulk data 500 times for each cell type to get a decent representation of the populations, and used the exact same procedure on the fibroblasts and immune cells. We validated this idea in simulated data (Supplementary Figure 5) and the pipeline worked as expected.

We agree that due to the limitations of the scRNA-seq technology, we cannot predict time for each individual cell and thus cannot directly measure or provide a score for phase heterogeneity in each cell type. But clearly, the pseudobulk approach and the incorporation of tauFisher with bootstrapping provides valuable insights into phase heterogeneity in multiple cell types, which would not be possible in bulk RNA-seq.

The claim that dermal immune cells have more heterogeneous phases than fibroblast cells requires experimental justification.

We agree with the reviewer that such an implication that immune cells have more heterogeneous phases than fibroblasts is based on computational analysis and an experimental validation is desired. However, our options are limited by technology constraints. We explored past journals, and investigation of circadian phase heterogeneity was only done *in vitro* using single-cell imaging tracking (Li et al., 2020). Such study *in vivo* focusing on multiple cell types is unprecedented given the complexities and difficulties of distinguishing cell types, labeling core clock genes, and continuously imaging mouse skin *in vivo* over a diurnal period.

We have explored multiple methods in an effort to validate tauFisher's heterogeneity prediction experimentally, including RNA fluorescence in situ hybridization and a time series of hybridization-based spatial RNA sequencing. Unfortunately, due to clock genes' low expression in cells, the methods did not yield results that can concretely validate or disprove our phase heterogeneity prediction. Given the constraints in obtaining experimental validation, we tested and verified the pipeline on simulated single cell data and found that indeed incorporating tauFisher with bootstrapping provides insights into phase heterogeneity (Supplementary Figure 5). Computational results remain the basis of our claim on phase heterogeneity until further experimental validation becomes feasible.

We now address this in the Discussion:

“Due to technological constrains, our claim on differences in phase heterogeneity between fibroblasts and immune cells relies on the computational analysis of the mouse skin scRNAseq and the simulated single-cell data.”

Reviewer #2 (Remarks to the Author):

Duan et al present a new algorithm, tauFisher, a "platform agnostic" circadian prediction algorithm. Given well annotated training data from a tissue of interest it aims to infer circadian phase in a new sample from the same tissue/cell type - even if obtained in a different platform or individual. They then apply this method to some newly collected single cell data and attempt to assess phase synchronization.

This (and related) problems are of increasing interest. I think this work is a significant contribution – I complement the authors.

Nonetheless I have some concerns. Much of this likely can be accommodated by the authors simply better describing the limits and assumptions of their method.

But I also have some more substantive concerns about the benchmarking - and their final use case

All in all - I think this work has value - and I suspect by concerns can be relatively easily addressed.

We appreciate the reviewer's comments and suggestions. We added in more figures and clarifying points to the manuscript to address the reviewer's concerns.

General Method:

Much of the improved robustness claimed by TauFisher appears to be attributed to its within sample normalization.

This seems like an interesting approach.

But I think there are several assumptions underlying this normalization. They should probably be more explicitly articulated/discussed

(1) In taking the differences between genes(i) and gene(j) in every sample - I think you are effectively assuming that those differences only depend on circadian time (and random noise) If there are systematic difference between the average levels of these genes between subjects (as might be expected in human data) - or between batches - this normalization would not address this? Similarly wide variation in amplitude among genes in people/subjects

(or course less concern with mouse data)

We agree with the reviewer that we assume that the differences between predictor genes, and thus the ordering of the differences, only depend on circadian time. This assumption stems from the fact that the core clock is a robust, interlocked transcription-translation feedback loop that ideally should generate consistent circadian patterns in healthy/control individuals. For example, in an undisturbed clock network in mouse skin, *Arntl* and *Dbp* should be antiphasic to each other, and the difference between these two genes should capture such a relationship. The ordering of differences between each possible pair of predictor genes provides more detailed information about where the gene network is at on the circadian clock. tauFisher uses this information to predict timestamps.

There are limitations of this assumption, which we addressed in the Discussion:

“...While tauFisher outperforms the ZeitZeiger and TimeSignatR in terms of both accuracy and RMSE in almost all benchmark datasets, its performance on human blood samples is subpar. Such performance can be attributed to greater human variability in expression patterns of the clock caused by hormone, stress, living style and diet. One assumption of tauFisher's within-sample normalization step is that the differences between predictor genes are solely dependent on circadian time, and this step only can remove uniform elevation or depression in expression values of clock genes. To further improve tauFisher's performance and flexibility on human data, it is worthwhile to incorporate a within-sample data centering step to address the differences caused by individual variability, while still making sure that only one test sample is needed to make a prediction. ...”

Additionally, we agree with the reviewer and do not expect this normalization step would address batch effects, as the normalization is done within each sample to allow circadian time prediction from just one test sample.

(2) Similarly, if you try to apply your model to a perturbation that may affect the amplitude or baseline expression of your genes (as many perturbations do) your training data on non-perturbed systems might not be appropriate?

For the case where the clock is disturbed and the relationships between the clock genes are altered compared to the ones in a normal/healthy clock, tauFisher trained on the normal/healthy data can still output a time prediction but the predicted time would reflect to the timescale of the healthy data, and thus not reflect the sample collection time of the disturbed tissue. By

comparing the sample collection time and the tauFisher-predicted time for the disturbed tissue, we can tell that the sample is disturbed but not how much it is disturbed.

On the other hand, tauFisher trained on samples from disturbed systems can be used to add time labels to samples collected from the same disturbed system.

We added in Supplementary Figure 2, and the following text in the Results section:

“After validating tauFisher’s performance on cross-platform, bulk-level, transcriptomic datasets collected from healthy/control mouse skin, we also tested it in disturbed systems. In the test groups of the time-restricted feeding study, food was only available to mice from ZT5 to ZT9 or ZT0 to ZT4, whereas mice usually feed during early nights (ZT12-ZT16) [6]. Skin collected from these two time-restricted feeding schedules showed disturbed circadian patterns with greatly attenuated amplitude and altered peaking times that are not directly correlated with the feeding times [6]. As the system is disturbed, the sample collection time no longer represents the internal circadian time of the tissue as it does in healthy tissue (training data). Consistent with the biological observations, tauFisher trained on control skin microarray data predicted time labels that are away from the test sample collection time, reflecting a disturbed system, and the predictions are not coupled with time-restricted feeding schedules (Supplementary Figure 2). tauFisher’s prediction when trained on control/healthy samples, however, can only tell whether the test system is disturbed or not, and does not provide a measurement of how much the system is disturbed.

We also trained the tested tauFisher within the disturbed systems. Within each of the two time-restricted feeding schedules, we performed leave-one-out cross validation by reserving each sample for testing and using the remaining samples for training. tauFisher produced high accuracy (feeding ZT5-ZT9: accuracy = 0.875; feeding ZT0-ZT4: accuracy = 1) and low RMSE (feeding ZT5-ZT9: RMSE = 2.236; feeding ZT0-ZT4: RMSE = 1.061) for both disturbed systems (Supplementary Figure 2). The fact that tauFisher trained on samples collected from a disturbed system can add time labels to samples from the same disturbed system suggests that robust correlations between diurnal genes still exist in the disturbed system, and such relationships are different from the ones in control/healthy individuals.”

We also added text discussing using disturbed samples for training and testing in the Discussion:

“Additionally, using tauFisher trained on healthy/control data to predict a time for data collected from a circadian-disturbed system only maps the disturbed test onto the timescale of the healthy samples. If one prefers to project a test data onto the timescale of the diseased samples, a time series of transcriptomic data from diseased individuals is required. However, the circadian pattern is dampened in many diseases including cancer [51–56], making the expression data similar overtime and thus more difficult to distinguish the time points. Although tauFisher showed promise in feeding-disturbed systems in skin and simulated dampened systems (Supplementary Figure 2 and 5), it would be valuable to further validate tauFisher on such datasets as they become available in future studies.”

(3) In assuming that the rank differences between genes i, j are roughly comparable between platforms - you are assuming that all measures all roughly scale the same amount - no? If the scaling factor for *Per2* from Array \rightarrow Seq is X 6, *Arntl* is x2 and *CRY1* is X 10 - the rank ordering of differences will be change.

We do not assume that the same scale factor applies to all genes when we compare between different platforms but the observation that tauFisher performs well when trained and tested on different platforms suggests that the relationships between the clock genes captured by different platforms are similar. This is expected as the clock network is very robust and the core clock genes that are antiphasic to each other, for example *Arntl* and *Dbp* should show antiphasic expression patterns no matter what assay platform is used. We have also included the ranking data frame calculated from mouse skin microarray data (train) and mouse skin scRNAseq pseudobulk (test) in the supplementary files (Supplementary Table 7 and 8) as examples if the reviewer is interested.

#####

Benchmarking

(1) In terms of the mouse tissue benchmarking Liver and Kidney (and then Lung) had the most robust rhythms in Zhang et al

(at least in terms of numbers of cycling genes - and my guess is in amplitude)

SCN is clearly one of the most robust brain tissues.

Could you evaluate tau fisher on the non-scn brain tissues in the Zhang data...

Help readers see how it does on "weaker tissues."

We evaluated tauFisher's performance on bulk RNAseq and microarray data from the cerebellum and brainstem in Zhang et al., and the results are now included in Figure 2. Similar to its performance in other tissues, tauFisher performs the best for microarray and bulk RNAseq data collected from the two brain regions.

We also updated the Results section:

"tauFisher achieved the highest accuracy for eleven of twelve benchmark datasets; ten using predictor genes found by JTK Cycle [36] and one using Lomb-Scargle [37] (Figure 2; Supplementary Table 2). ZeitZeiger could not predict the time for several iterations due to linearly dependent basis vectors. Particularly, in the kidney and liver bulk RNAseq datasets and the brainstem and cerebellum datasets, ZeitZeiger failed to predict the time for all 100 iterations. Interestingly, whether ZeitZeiger ran into linear dependency issues depended on data type, as it ran successfully for most of the microarray data and failed to make predictions for most of the bulk RNAseq data. While two test samples were provided to TimeSignalR per prediction, it did not attain the highest accuracy in any of the datasets and only achieved the lowest root mean squared error (RMSE) in two of the twelve datasets."

(2) In terms of the blood data - I could not tell if the individual subjects contributed to both the training and testing samples in each run

Or if you made sure to train and test on disjoint subjects (rather than disjoint samples). It should be done that way. If it was already - great. If not pleas redo

We redid the benchmark on human blood data so that the methods train and test on disjoint subjects. The performance of all three methods are subpar for both human blood microarray and bulk RNAseq data.

We think such performance can be attributed to human variability and added the following text to the Discussion section:

“While tauFisher outperforms the ZeitZeiger and TimeSignatR in terms of both accuracy and RMSE in almost all benchmark datasets, its performance on human blood samples is subpar. Such performance can be attributed to greater human variability in expression patterns of the clock caused by hormone, stress, living style and diet. One assumption of tauFisher's within-sample normalization step is that the differences between predictor genes are solely dependent on circadian time, and this step only can remove uniform elevation or depression in expression values of clock genes. To further improve tauFisher's flexibility and performance on human data, it is worthwhile to incorporate a within-sample data centering step to address the differences caused by individual variability, while making sure that only one test sample is needed to make a prediction.”

(3) Also in terms of blood data - this seems like a good venue to test your cross-platform abilities

Can you provide data showing what happens when you use the model trained on Arnardottier et al on the Braun data (and vice versa)

I think this would help assess tauFisher in some more realistic use cases - and easy to do given your work already!

Since tauFisher's performance when trained and tested within the microarray or bulk RNAseq data is already subpar, we do not think it is worthwhile to include the human blood cross platform training-test results in the manuscript. We attach the human blood cross platform results below in case the reviewer is interested.

I don't think you ever define "Accuracy"

I gather you are using as tauFischer is a classifier and provides discrete labels rather than continuous times - and is just percent assigned to correct label?

But I couldn't find in manuscript.

We call a prediction correct when it is within 2-hour of the true timestamp. Accuracy is then calculated as the percentage of accurate predictions.

We defined accuracy in the result section:

“We define a prediction within two hours of the true time to be correct. Using other time ranges to define correctness minimally change the benchmark outcome (Supplementary Table 1).”

We also updated the figures (Figure 3 and Supplementary Figure 3) to distinguish and define accurate predictions for better clarification.

For the data that had 2,3, or 4 hour resolution, was the TauFischer multivariate logistic model limited to those possibilities? Or did you keep 24-hour resolution - and only count correct if it got the exact right time?

tauFischer works for datasets collected with different time internals as the functional data analysis step fills in the missing time points, and we keep that 24-hour resolution in both training and testing in this manuscript. Because the functional data analysis step produces a smooth, continuous curve, users can also define the resolution they want to use in the pipeline (e.g. half hour, 2 hours, 4 hours etc.).

We mentioned the purpose of using functional data analysis in the introduction:

“The training part of the pipeline ... (2) curve fitting using functional data analysis to fill in the missing time points and to decrease noise in the training data...”

We also discussed the purpose of using functional data analysis in the Discussion:

“Functional data analysis for the training data enables tauFischer to remove minor noise, smooth the time expression curves, and generate the expression data between the sampled time points.”

We call a prediction correct when it is within 2-hour of the true timestamp. Accuracy is then calculated as the percentage of accurate predictions.

We defined accuracy in the Results:

“We define a prediction within two hours of the true time to be correct. Using other time ranges to define correctness minimally change the benchmark outcome (Supplementary Table 1).”

We also updated the figures (Figure 3 and Supplementary Figure 3) to distinguish and define accurate predictions for better clarification.

On blood TauFischer (and the other methods) seem to have a <50% accuracy, and an RMSE error near 5-6 hours (+/- 5 hour window is a pretty big range)

We think human variability causes tauFischer to not perform as well on human blood data. We added a discussion about tauFischer’s performance on human blood samples in the Discussion:

“While tauFischer outperforms the ZeitZeiger and TimeSignatR in terms of both accuracy and RMSE in almost all benchmark datasets, its performance on human blood samples is subpar. Such performance can be attributed to greater human variability in expression patterns of the clock caused by hormone, stress, living style and diet. One assumption of

tauFisher's within-sample normalization step is that the differences between predictor genes are solely dependent on circadian time, and this step only can remove uniform elevation or depression in expression values of clock genes. To further improve tauFisher's flexibility and performance on human data, it is worthwhile to incorporate a within-sample data centering step to address the differences caused by individual variability, while making sure that only one test sample is needed to make a prediction."

You mention the puzzling performance attributed to TimeSignatR - it makes me wonder if it was used correctly?

As you note in you note in the discussion the "2 point" version did worse than the 1 point?

Could you explain what you mean when you say the original paper used the test data in training?

I am not trying to cause a feud! So if you prefer to leave out of paper that is fine!

But it would help me feel more confident that assessment was done fairly

In the pipeline of TimeSignatR, a step called 2-point calibration was applied to the test samples as a data processing step. In this step, test samples that are around 12 hours apart (antipodal samples) were paired up. Then, a mean is calculated from the expression values of the two samples, and the calibration step subtracts the mean from the test sample. Finding a pair of test samples that are around 12 hours apart requires knowledge of the time labels of the test samples, and thus we found it unfair to benchmark ZeitZeiger and tauFisher against TimeSignatR's original pipeline.

We quoted and discussed how we understood the TimeSignatR vignette in detail in the Supplementary File. We also did an experiment where we hid the true time labels of the test samples by labeling them with randomly generated time labels. TimeSignatR's results in this experiment are worse (Supplementary File Section: Discussion on TimeSignatR), confirming the fact that TimeSignatR uses time labels of test data to process the test data in the pipeline.

Although we are sure that we read about TimeSignatR being able to run when provided with only one sample in one of the vignette websites (and we even still have the code), that website is no longer available, and the TimeSignatR paper states that it requires at least two samples. We modified our manuscript so that when benchmarking TimeSignatR, it is always provided with two samples. However, since the original TimeSignatR pipeline requires knowledge of test time labels to find a second sample that is around 12 hours away while tauFisher and ZeitZeiger do not use time labels of test data in the pipelines, we modified the TimeSignatR pipeline to make the benchmark fair. For TimeSignatR's two-point calibration step, we randomly pulled a second sample from the test sample pool so that the time labels of the test data are not used by TimeSignatR.

We updated our Results:

"We compared tauFisher to the current state-of-the-art methods: ZeitZeiger [28] and a modified version of TimeSignatR (Methods, Supplementary Files) [30]."

We updated our Methods:

"For TimeSignatR, the original pipeline requires knowledge of the time labels of test samples to find antipodal samples (samples that are around 12 hours apart) in the 2-point calibration step, while tauFisher and ZeitZeiger do not use test time labels at all. To

make the benchmark fair, we modified the TimeSignatR pipeline so that the second sample in the 2-point calibration step is randomly selected from the test sample pool. In this way, TimeSignatR does not use timestamps of test data in data processing.”

We updated our Discussion:

“TimeSignatR performed less well in the benchmark, even though it uses two samples. We reported worse general performance for TimeSignatR than reported in [30]. The main cause can be attributed to our modification of their 2-point calibration step (Methods). In the original TimeSignatR pipeline, time labels of test samples are available to TimeSignatR as it needs to pick out antipodal test samples that are around 12 hours apart in the calibration step (Supplementary File). As tauFisher and ZeitZeiger do not utilize time labels of test samples in the processing of test data, to make a fair comparison of their performance in predicting circadian time, we modified TimeSignatR's code so that the pipeline no longer uses time labels of test samples and it randomly pulls a second sample from the test sample pool to do the 2-point calibration.”

I don't know the red stars are on the plots

The red stars were outliers. To make Figure 2 less crowded, we have removed the red stars as points outside of the whiskers are outliers.

You mention a new method Tempo that estimates prediction uncertainty

Is this easy to benchmark? - this is not required - you have already tested 2 other methods which I take as a good faith effort.

But if you can't - I wonder - there is a wide variety in the prediction accuracy that you do show.

Can you guide the user as to how to estimate the fit confidence of Tau Fishcer in a new application

We thank the reviewer for the suggestion of benchmarking Tempo. However, we decided not to benchmark Tempo as it is specifically designed to infer circadian phase for each individual cell in scRNAseq, whereas ZeitZeiger, TimeSignatR and tauFisher are primarily designed to add time labels to bulk/pseudobulk data. As the goals and scales of Tempo are different from the methods we benchmarked in the manuscript, we prefer to evaluate Tempo in our future studies so that the current manuscript is more focused.

The current version of tauFisher can print out the multinomial probability (plots below as an example, not included in the manuscript), but unfortunately it cannot perform a full uncertainty quantification like Tempo.

We added a discussion of tauFisher's limitations in the Discussion:

“... as they become available in future studies. Additionally, since uncertainty quantification may be particularly important when testing in disturbed systems, tauFisher can be improved to output confidence scores. Finally, ...”

#####

New Application

Most of the analysis of the new scRNA-seq dermal skin data was of course done without tauFisher - as collection time was known.

The interesting analysis that used tau-fischer was based on using TauFischer to try to assess if the reduced amplitude in immune cells as opposed to fibroblasts was due to desynchrony vs a lower amplitude in each cell

The authors take subsamples of the single cell pseudobulk - and assess the variability in their phase estimates /do a bootstrap analysis.

They find that the estimates are more spread out in the immune cells and relate this to decreased synchronization.

(1) It seems to me that if you assumed significant random variation in the DC offset term (the D component of your sinusoidal variation model for each gene) this would also give you more variability in your phase estimate. Why do you assume variability in phase among cells is a better explanation than simply more variation in mean expression among these cells?

We observed dampened amplitudes in the expression of core clock genes in immune cells at a pseudobulk level, and our goal was to understand the potential cause behind this observation. Variations in the D components (vertical shifts) in expression value in individual cells does not lead to dampened amplitudes at the pseudobulk level and was thus not considered.

We updated Supplementary Figure 5. We simulated single cell data with vertical shifts in their circadian pattern (Supplementary Figure 5c) and calculated their pseudobulk data to demonstrate that variations in mean expression does not cause dampened amplitude at the pseudobulk level (Supplementary Figure 5d).

We also added the following text to the Results:

“Note that variations of mean expression in single cells (vertical shifts of expression curves) do not cause dampened amplitudes at the pseudobulk level (Supplementary Figure 5), so this scenario is not considered in the following analysis.”

(2) A lesser point - my understanding is that the measurement of more lowly expressed transcripts are also innately more variable in scRNAseq - so wouldn't you expected to see more variable measurements anyway?

We added a supplementary figure to compare the distributions of expression levels of core clock genes in fibroblasts and immune cells at different time points (Supplementary Figure 4). For each cell, the expression level is calculated and normalized as the transcript count divided by the total number of reads in that cell and then times 10000. While for *Per1* the expression range is greater in immune cells than fibroblasts at all time points, there is no consistent pattern for the other clock genes (Supplementary Figure 4a).

Because the clock gene expression distributions are greatly skewed to 0 due to high sequencing dropout rate, we use interquartile range (the difference between the upper quartile and the lower quartile) to measure the spread of the distributions (Supplementary Figure 4b). We see that in general the expression of core clock genes is more spread out in fibroblasts than in immune cells, although at many timepoints the interquartile range values are 0 for both fibroblasts and immune cells.

We added a sentence discussing the expression distribution of the core clock genes in the Results section:

“In general at the single cell level, the expression ranges of the core clock genes are similar in the two cell types, and the measurements of the clock genes in fibroblasts are more variable (Supplementary Figure 4). To compare the core clock in fibroblasts and immune cells, we computed and normalized the pseudobulk data ...”

#####

For the enrichment analysis - what was the background list used? Transcripts expressed in the cell - or the whole genome?

We updated the Methods section:

“For rhythmic genes with $JTK_pvalue < 0.01$ in the dermal fibroblasts or immune cells, we used the hypergeometric test ... We used the transcripts expressed in the cells as the background gene list in the enrichment analysis. False discovery rate...”

#####

Discussion

Some of the points you rightly emphasize are in eventual application for circadian medicine. Like learning cardiac time given myocardial injury data. Similarly adding "time" information to public data repositories. But I think you should probably emphasize that tauFisher would need training data for that human tissue - and likely training data unique to any disease state.

We added the following text in the Discussion:

“Additionally, using tauFisher trained on healthy/control data to predict a time for data collected from a circadian-disturbed system only maps the disturbed test onto the timescale of the healthy samples. If one prefers to project a test data onto the timescale of the diseased samples, a time series of transcriptomic data from diseased individuals is required. However, the circadian pattern is dampened in many diseases including cancer [51–56], making the expression data similar overtime and thus more difficult to distinguish the time points. Although tauFisher showed promise in feeding-disturbed systems in skin and simulated dampened systems (Supplementary Figure 2 and 5), it would be valuable to further validate tauFisher on such datasets as they become available in future studies.”

Also that the +/- 5 hour window they give (RMSE) for the human blood data - might suggest that human variability remains significant obstacle. I know the authors know this - but given the discussion points it should be explicit.

We added the following text in the Discussion:

“While tauFisher outperforms the ZeitZeiger and TimeSignatR in terms of both accuracy and RMSE in almost all benchmark datasets, its performance on human blood samples is subpar. Such performance can be attributed to greater human variability in expression patterns of the clock caused by hormone, stress, living style and diet. One assumption of tauFisher's within-sample normalization step is that the differences between predictor genes are solely dependent on circadian time, and this step only can remove uniform elevation or depression in expression values of clock genes. To further improve tauFisher's flexibility and performance on human data, it is worthwhile to incorporate a within-sample data centering step to address the differences caused by individual variability, while making sure that only one test sample is needed to make a prediction.”

Also some more focused attention to where you think the weakness of tau fisher are warranted.

We added a paragraph in the Discussion to discuss some of the improvements tauFisher can make, including the aspects mentioned above:

“Despite its unique circadian time prediction ability, tauFisher can be improved in several ways. While tauFisher outperforms the ZeitZeiger and TimeSignatR in terms of both accuracy and RMSE in almost all benchmark datasets, its performance on human blood samples is subpar. Such performance can be attributed to greater human variability in expression patterns of the clock caused by hormone, stress, living style and diet. One assumption of tauFisher's within-sample normalization step is that the differences between predictor genes are solely dependent on circadian time, and this step only can remove uniform elevation or depression in expression values of clock genes. To further

improve tauFisher's flexibility and performance on human data, it is worthwhile to incorporate a within-sample data centering step to address the differences caused by individual variability, while making sure that only one test sample is needed to make a prediction. Additionally, using tauFisher trained on healthy/control data to predict a time for data collected from a circadian-disturbed system only maps the disturbed test onto the timescale of the healthy samples. If one prefers to project a test data onto the timescale of the diseased samples, a time series of transcriptomic data from diseased individuals is required. However, the circadian pattern is dampened in many diseases including cancer [51–56], making the expression data similar overtime and thus more difficult to distinguish the time points. Although tauFisher showed promise in feeding-disturbed systems in skin and simulated dampened systems (Supplementary Figure 2 and 5), it would be valuable to further validate tauFisher on such datasets as they become available in future studies. Additionally, since uncertainty quantification may be particularly important when testing in disturbed systems, tauFisher can be improved to output confidence scores. Finally, while tauFisher accurately adds timestamps to unlabeled scRNAseq data and can predict circadian time for pseudobulk data generated from a group of cells in scRNAseq data, it cannot overcome the high sequencing dropout rate in scRNAseq and thus cannot predict circadian time at a single cell level. Future work could focus on incorporating an imputation step into tauFisher to infer expression values of predictor genes. This step may not sacrifice computation and time efficiency greatly, as tauFisher only need around 15 genes to make predictions.”

#####

Title:

A fairer title might be

"tauFisher accurately predicts circadian time from a single sample of bulk or pseudobulk transcriptomic data"

You never really do assign phase to a single-cell ..

We thank the reviewer for this suggestion and we have now changed the title to “tauFisher accurately predicts circadian time from a single sample of bulk or single cell pseudobulk transcriptomic data” to avoid confusion.

Finally, I must admit I have not tested code.

It seems to all be there- and it looks pretty - but I have not run it myself.

I asked a student to run it (and he was able to)

We thank the reviewer for the feedback, and we are glad to hear that the vignette ran smoothly.

Reviewer #3 (Remarks to the Author):

TauFisher is presented as an algorithm to predict circadian time from a single experiment transcriptomes, including from single-cells. The method has a number of interesting features: (1) it is applicable independent of the transcriptome platform and apparently performs well in a cross platform setting; (2) does not require the training data to be a complete time series; (3) it uses comparisons of clock genes within a sample; (4) it can work for single cell data. Overall the presentation of some of the analyses needs to be improved and in general the methods need to be better explained.

We appreciate the reviewer's comments and suggestions. We have improved the presentation of the method and the results accordingly.

Results 2.1

1. The method part with the difference matrix is not well explained, particularly the scaling. This matrix should be shown and compared across datasets.

We thank the reviewers for raising this issue. We have updated the Methods section to be clearer on the calculation of the difference matrix and the scaling:

“For each time point, tauFisher generates all possible pairings of the selected predictor genes and calculates the differences between the two genes' FDA-smoothed expression stored in matrix Y . The resulting matrix retains differences between Gene a and Gene b (Gene a - Gene b) as well as between Gene b and Gene a (Gene b - Gene a). Then within each time point, the differences calculated from gene pairs are scaled to be between 0 and 1 using the `rescale` function in R package `scales`. This way, 0 represents the minimum difference value and 1 represents the maximum. The formula is $(\text{value} - \text{min})/(\text{max} - \text{min})$.”

We have now included the rescaled difference matrix calculated from mouse skin microarray data (train) and mouse skin scRNAseq pseudobulk (test) in the supplementary files (Supplementary Table 7 and 8) as an example. We are not able to show the difference matrices for the datasets involved in the benchmark (Figure 2) since there are 100 training-testing partition outcomes for each of the data and showing the matrices for each partition is not feasible. If users are interested, the difference matrices and the intermediary steps can be output to the users at any given point (see vignette).

How does it look in broken clock conditions?

For the case where the clock is disturbed and the relationships between the clock genes are altered compared to the ones in a normal/healthy clock, tauFisher trained on the normal/healthy data can still output a time prediction but the predicted time would reflect to the timescale of the healthy data, and thus not reflect the sample collection time of the disturbed tissue. By comparing the sample collection time and the tauFisher-predicted time for the disturbed tissue, we can tell that the sample is disturbed but not how much it is disturbed.

On the other hand, tauFisher trained on samples from disturbed systems can be used to add time labels to samples collected from the same disturbed system.

We added in Supplementary Figure 2 to the Supplements, and the following text in the Results section:

“After validating tauFisher's performance on cross-platform, bulk-level, transcriptomic datasets collected from healthy/control mouse skin, we also tested it in disturbed systems. In the test groups of the time-restricted feeding study, food was only available to mice from ZT5 to ZT9 or ZT0 to ZT4, whereas mice usually feed during early nights (ZT12-ZT16) [6]. Skin collected from these two time-restricted feeding schedules showed disturbed circadian patterns with greatly attenuated amplitude and altered peaking times that are not directly correlated with the feeding times [6]. As the system is disturbed, the sample collection time no longer represents the internal circadian time of the tissue as it does in healthy tissue (training data). Consistent with the biological observations, tauFisher trained on control skin microarray data predicted time labels that

are away from the test sample collection time, reflecting a disturbed system, and the predictions are not coupled with time-restricted feeding schedules (Supplementary Figure 2). tauFisher's prediction when trained on control/healthy samples, however, can only tell whether the test system is disturbed or not, and does not provide a measurement of how much the system is disturbed.

We also trained the tested tauFisher within the disturbed systems. Within each of the two time-restricted feeding schedules, we performed leave-one-out cross validation by reserving each sample for testing and using the remaining samples for training. tauFisher produced high accuracy (feeding ZT5-ZT9: accuracy = 0.875; feeding ZT0-ZT4: accuracy = 1) and low RMSE (feeding ZT5-ZT9: RMSE = 2.236; feeding ZT0-ZT4: RMSE = 1.061) for both disturbed systems (Supplementary Figure 2). The fact that tauFisher trained on samples collected from a disturbed system can add time labels to samples from the same disturbed system suggests that robust correlations between diurnal genes still exist in the disturbed system, and such relationships are different from the ones in control/healthy individuals."

We also added text discussing using disturbed samples for training and testing in the Discussion:

"Additionally, using tauFisher trained on healthy/control data to predict a time for data collected from a circadian-disturbed system only maps the disturbed test onto the timescale of the healthy samples. If one prefers to project a test data onto the timescale of the diseased samples, a time series of transcriptomic data from diseased individuals is required. However, the circadian pattern is dampened in many diseases including cancer [51–56], making the expression data similar overtime and thus more difficult to distinguish the time points. Although tauFisher showed promise in feeding-disturbed systems in skin and simulated dampened systems (Supplementary Figure 2 and 5), it would be valuable to further validate tauFisher on such datasets as they become available in future studies."

Results 2.2

2.The presentation of Figure 2 is very busy, i would suggest removing all the p-values. The red stars are not described, what are those?

We agree with the reviewer that Figure 2 contains a lot of information. However, we do think that it is necessary to include the asterisks to indicate p-values, as it is important to know whether the performance measures for the pipelines are significantly different from each other. To make the figure less cluttered, we removed the "ns" for non-significant comparisons and kept the asterisks for the significant ones.

The red stars were outliers. To make Figure 2 less crowded, we have removed the red stars as points outside of the whiskers are outliers.

3.It seems that TauFisher performs worse than Zeitzeiger in many cases. Was this expected?

According to the benchmark results presented in Supplementary table 2, tauFisher outperformed ZeitZeiger in terms of accuracy and RMSE in eleven of the twelve benchmark datasets. Additionally, while ZeitZeiger was often not able to make predictions due to linear

dependency issues for the bulk RNAseq data, tauFisher ran successfully on all of the benchmark datasets from different organs and assay platforms.

To clarify this, we have discussed the benchmark results in the Results section:

“tauFisher achieved the highest accuracy for eleven of twelve benchmark datasets; ten using predictor genes found by JTK Cycle [36] and one using Lomb-Scargle [37] (Figure 2; Supplementary Table 2). ZeitZeiger could not predict the time for several iterations due to linearly dependent basis vectors. Particularly, in the kidney and liver bulk RNAseq datasets and the brainstem and cerebellum datasets, ZeitZeiger failed to predict the time for all 100 iterations. Interestingly, whether ZeitZeiger ran into linear dependency issues depended on data type, as it ran successfully for most of the microarray data and failed to make predictions for most of the bulk RNAseq data. While two test samples were provided to TimeSignatR per prediction, it did not attain the highest accuracy in any of the datasets and only achieved the lowest root mean squared error (RMSE) in two of the twelve datasets.”

4. Taufisher seems to be sensitive to the seed genes. How is the performance dependent on the number of seed genes?

We thank the reviewer for bringing up this important question. To investigate tauFisher's performance when using different numbers of genes, we performed a sensitivity analysis. Besides the core clock genes with period length of 24 hours, additional numbers of diurnal genes were included to be predictor genes in the tauFisher pipeline. We did the sensitivity analysis for three cases: 1. Training on mouse SCN bulk RNAseq and testing on mouse SCN scRNAseq pseudobulk; 2. Training on mouse skin microarray and testing on mouse skin bulk RNAseq; and 3. Training on mouse skin microarray and testing on mouse dermal skin scRNAseq pseudobulk (Supplementary Figure 3d). For the three cases, the addition of the first few predictor genes improved tauFisher's performance in terms of both accuracy and RMSE but prediction became worse if too many genes were used. We chose to use 10 diurnal genes in addition to the core clock genes with 24-hour period length throughout the manuscript. Although this does not guarantee the best possible performance, it is a reasonable and safe choice across different datasets (Supplementary Figure 3d).

We have updated the Methods section:

“... then selects the top ten statistically significant genes with a 24-hour period length. Using different numbers of diurnal genes affects the prediction outcome. Although choosing the top ten does not guarantee the best performance, it is a safe and reasonable choice across different datasets (Supplementary Figure 3).”

5. The accuracy in blood is always quite low, can the authors comment on it?

We think human variability causes tauFisher to not perform as well on human blood data. We have added a discussion about tauFisher's performance on human blood samples in the Discussion:

“...While tauFisher outperforms the ZeitZeiger and TimeSignatR in terms of both accuracy and RMSE in almost all benchmark datasets, its performance on human blood samples is subpar. Such performance can be attributed to greater human variability in

expression patterns of the clock caused by hormone, stress, living style and diet. One assumption of tauFisher's within-sample normalization step is that the differences between predictor genes are solely dependent on circadian time, and this step only can remove uniform elevation or depression in expression values of clock genes. To further improve tauFisher's flexibility and performance on human data, it is worthwhile to incorporate a within-sample data centering step to address the differences caused by individual variability, while making sure that only one test sample is needed to make a prediction ...”

6. Which were the selected genes in the different cases? Are the genes the same for the three methods? What if the same genes are used in all three methods?

Unfortunately, we are not able to display the selected genes related to the benchmark result shown in Figure 2 as we generated 100 random training and test partitions for each of the datasets. Because the samples used for training are different in the partitions, the selected genes according to the training data will also be slightly different, meaning that we are looking at a list contains $100 \times 4 \times 8 \times \sim 20$ (number of partitions * number of feature selection methods from the three pipelines * number of benchmark datasets * approximate number of selected genes in each pipeline) = ~ 64000 genes. The genes can be output to users if they are interested (see vignette). We went through the genes used in the three methods and found that the gene sets are different, with most of the overlapping genes being clock genes.

Since gene selection is part of the pipeline for ZeitZeiger and TimeSignatR, we assume that the genes picked in those two pipelines work the best for themselves and thus it is only fair to run these two methods with their built-in gene selection procedure in the benchmark.

We displayed the predictor genes selected by tauFisher in Figure 3 for the cross-platform predictions. tauFisher also outputs selected predictor genes to users (vignette).

7. It would be useful to show RMSE and Accuracy on training and testing sets separately.

We thank the reviewer for the suggestion. We now included RMSE and Accuracy of tauFisher on two training datasets, mouse SCN bulk RNAseq (Supplementary Figure 3a) and mouse skin microarray (Supplementary Figure 3b), as examples.

Results 2.3

8. Fig3A. Axis labels are missing, what are the units? It should be shown in log₂ scale as the fits were done in log-scale. In general, the authors should make sure that there is no mix-up of linear and log scale.

For the original Figure 3a and now Figure 3a,c,f, the labels are now updated. For all bulk RNAseq/microarray data in these panels, the y axes are in reads per kilobase per million, and for the scRNAseq pseudobulk data, the y axes are in counts. We are certain that there is no mix-up of linear and log scale.

We prefer to show the raw training and test data as it is input into the tauFisher pipeline in the main figure. Since log₂ transformation is an intermediary step, we added the plots for the log₂-transformed data into Supplementary Figure 1.

9. A scatter plot of predicted vs true sample time should be plotted

We have now updated all of our prediction outcome radar plots to scatter plots (Figure 3, Supplementary Figure 3), and we agree with the reviewer that the scatter plots display the data better.

10. The right circular plots are difficult to interpret and not informative

As mentioned above, we now have updated all of our prediction outcome radar plots to scatter plots (Figure 3, Supplementary Figure 3), and we agree with the reviewer that the scatter plots display the data better.

Results 2.4

11. These sentences are very confusing: “The test data appears to be noisier since it is not normalized. tauFisher does not require the test data to be normalized as the within-sample normalization step is part of the pipeline. Explain better.

We have edited the sentence to make it clearer:

“The raw input test data appeared to be noisier as it was not normalized by the total number of reads in each sample. tauFisher does not require the data to be preprocessed before input into the pipeline, as within-sample normalization is an intermediary step.”

12. A scatter plot would be more informative and easier to read than Table 2.

We thank the reviewer for this suggestion. We have replaced the original table with a scatter plot.

13. It is not so surprising that pseudo-bulk patterns are similar to bulk, this is most typically the case in scRNA-seq

We agree with the reviewer that the scRNAseq pseudobulk data often display similar patterns to bulk RNAseq data, but we still want to verify this assumption.

We needed to verify the consistency between bulk RNAseq data and scRNAseq pseudobulk data because the experimental protocols to obtain these two types of data are different, and such differences could be important in circadian studies. In bulk RNAseq experiments, the tissue samples are usually snap-frozen in liquid nitrogen as soon as they are excised from the body. The samples are then ground and lysed for RNA extraction. The whole process usually takes around 30 minutes. One can also skip the snap-freezing step and obtain bulk RNAseq from fresh tissue. In contrast, in scRNAseq experiments, the tissue samples go through multiple incubations in different digestion solutions at 37C until live cells are dissociated from each other to form a single cell suspension. The suspension is then usually kept on ice for dead cell removal. The whole process takes a few hours. As temperature can affect the clock (Buhr et al., 2010) and the preprocessing of the samples before sequencing takes different amount of time, it is necessary to verify that a time series of bulk RNAseq and a time series of scRNAseq pseudobulk data are consistent with each other. One way to verify this is to overlay the bulk transcriptomics data with the pseudobulk data. For the SCN data, such consistency is shown in (Wen et al., 2020). For the skin, we demonstrate such consistency in Figure 4a in this manuscript. The fact that tauFisher trained on bulk RNAseq/microarray data can accurately predict timestamps for scRNA-seq pseudobulk data reassures such consistency. These

observations also suggest that clock gene expression represents the clock at the time of sample isolation in both bulk RNA and scRNA extractions.

To highlight the importance of these considerations, we added the following text to the Discussion:

“While it is usually assumed that bulk RNAseq data are consistent with pseudobulk data calculated from scRNAseq data, it is necessary to verify this assumption in circadian studies. The experimental settings for bulk and single cell RNA extraction are different in terms of digestion duration and temperature, a factor that can alter the clock [4]. In this paper, we verify the consistency of circadian patterns in the two types of data by overlaying the expression of core clock genes (Figure 4a); tauFisher’s successful bulk-to-pseudobulk predictions (Figure 3) reassure such consistency. However, tauFisher is the only method that successfully predicted the circadian time for the pseudobulk data (Figure 3) despite the consistency between bulk RNAseq/microarray data and scRNAseq pseudobulk data. The robustness in performance despite drastically different assay methods and experimental setups suggests that tauFisher captures and extracts the underlying biological correlations in gene expressions while minimizing the effects of the noise and variability introduced by subjects and technology.”

Results 2.5/2.6

14. While these sections are potentially interesting, these analyses seem unrelated with the rest of the paper (tauFisher). It seems like these sections should go in a separate manuscript, or they will not be noticed inside a methods paper.

We appreciate the reviewer’s concern that the biological data may not get the exposure that it deserves. We, however, would prefer to keep these findings in this manuscript as we believe including biological insights will enhance the impact of the paper. In addition, these data allow us to highlight the use of tauFisher to predict features of the clock in different cell types within a complex tissue. We plan to emphasize the biological findings in future review article(s) that would point readers to the primary research paper.

Results 2.7

15. The bootstrapping procedure is quite complex and should be better described.

We edited the related text to make it more organized and clearer:

“Since the heterogeneity of a set of heterogeneous clocks should be captured at any given time point, we performed the analysis within each time point. The workflow involves the following steps: (1) Trimming the scRNAseq data so that the expression matrix only includes the predictor genes identified in the training data and the cells labeled to be the interested cell types; (2) Randomly sampling the same number of cells for each cell type to remove potential bias caused by different cell numbers, and summing the transcript counts in the pulled cells for each gene to create a pseudobulk dataset; (3) Repeating the random sampling process (step 2) with replacement many times to create pseudobulk replicates for each cell type; and (4) Predicting circadian time labels for the pseudobulk replicates using tauFisher. The idea is that if the cells harbor synchronous clocks, the pseudobulk replicates calculated from different rounds of sampling will be similar. In this case, the distribution of predicted time labels will be more concentrated. On the other hand, if the cells harbor heterogeneous clocks, the pseudobulk replicates calculated from the cells in different rounds of sampling will differ

depending on which cells are pulled. The distribution of the prediction outcome in this case will be wider.”

16. The authors should clarify the argument behind “indicating that immune cells have a more heterogeneous clock phase than fibroblasts.”

To compare phase heterogeneity between cell types, we used the pseudobulk approach applied to individual cell types instead of to the entire sample. The idea is that, if we pull from a group of synchronous clocks (for example, all cells are at ZT10), then the pseudobulk calculated from the pulled cells will be very similar in the 500 rounds, and thus the prediction outcome will have a very narrow distribution. On the other hand, if we pull from a group of heterogeneous clocks (for example, 20% at ZT10, 20% at ZT12, 50% at ZT23 and 10% at ZT6), the pseudobulk data calculated from the cells in each round of pulling will differ depending on which cells are pulled. The distribution of the prediction outcome in this case will be wide.

We added the following text to the Results section:

“The idea is that if the cells harbor synchronous clocks, the pseudobulk replicates calculated from different rounds of sampling will be similar. In this case, the distribution of the predicted time labels will be more concentrated. On the other hand, if the cells harbor heterogeneous clocks, the pseudobulk replicates calculated from the cells in different rounds of sampling will differ depending on which cells are pulled. The distribution of the prediction outcome in this case will be wider.”

To ensure that noise in data is not causing bias in the final conclusion, we randomly sampled cells to calculate pseudobulk data 500 times for each cell type to get a decent representation of the populations, and used the exact same procedure on the fibroblasts and immune cells. We validated this idea in simulated data and the pipeline worked as expected (Supplementary Figure 5).

We also edited our conclusion sentence:

“Indeed, the standard deviation of the prediction distribution is significantly greater for immune cells for five out of the six ZTs. This means that the bootstrapping pulled from a more heterogeneous population when sampling the immune cells, and thus implying that the clock phases are more heterogeneous in immune cells than in fibroblasts.”

Reference:

- Anafi, R. C., Francey, L. J., Hogenesch, J. B., & Kim, J. (2017). CYCLOPS reveals human transcriptional rhythms in health and disease. *Proceedings of the National Academy of Sciences of the United States of America*, 114(20), 5312–5317.
- Buhr, E. D., Yoo, S. H., & Takahashi, J. S. (2010). Temperature as a universal resetting cue for mammalian circadian oscillators. *Science (New York, N.Y.)*, 330(6002), 379–385.

- Wen, S., Ma, D., Zhao, M., Xie, L., Wu, Q., Gou, L., Zhu, C., Fan, Y., Wang, H., & Yan, J. (2020). Spatiotemporal single-cell analysis of gene expression in the mouse suprachiasmatic nucleus. *Nature neuroscience*, 23(3), 456–467.
- Li, Y., Shan, Y., Desai, R. V., Cox, K. H., Weinberger, L. S., & Takahashi, J. S. (2020). Noise-driven cellular heterogeneity in circadian periodicity. *Proceedings of the National Academy of Sciences of the United States of America*, 117(19), 10350–10356.

Reviewer #1 (Remarks to the Author):

I am satisfied with most answers to my previous questions except for this one regarding Figure 6b-c:

My previous question remains:

"The fact that the mean phase of immune cells is advanced compared to fibroblast at some ZT times but delayed at other ZT times (Figure 6b) indicates that the fluctuation of phases is more due to the estimation procedure itself rather than the biological differences."

In the revised paper, the authors stated that "Whether one cell type's circadian clock is ahead of the other is inconclusive (Figure 6c)." but statistically significant differences (p -value ≤ 0.0001) were found for most ZT times. ZT6 even has the opposite sign from other ZTs. Furthermore, the means of predicted time stamps of fibroblasts and immune cells deviate from real sample collection times at several ZTs, e.g. ZT10. This seems contradictory to the robustness of phase estimation claimed in the text.

The authors continued on to say "Indeed, the standard deviation of the prediction distribution is significantly greater for immune cells for five out of the six ZTs." but again statistically significant differences were found for standard deviation for most ZT times. The difference of SD at ZT6 is again opposite to the differences at other ZTs.

I also suggest the authors change the wordings to avoid over-statement in places such as:

"unprecedentedly, tauFisher trained on bulk sequencing data is able to accurately predict the circadian time of single-cell RNA sequencing (scRNAseq) pseudobulk data"

"However, tauFisher is the only method that successfully predicted the circadian time for the pseudobulk data"

Reviewer #2 (Remarks to the Author):

I thank the Authors for taking the time to try to thoroughly address my concerns. They have improved the paper.

However, as they clarified their benchmarking methods, it is now clear they have been quite unfair in the benchmarking/analysis of the TimeSignatR method. This is now (unfortunately) a major concern.

Major:

The text added to the manuscript (in response to my question about TimeSignatR) is given below:

"The main cause can be attributed to our modification of their 2-point calibration step (Methods). In the original TimeSignatR pipeline, time labels of test samples are available to TimeSignatR as it needs to pick out antipodal test samples that are around 12 hours apart in the calibration step (Supplementary File). As tauFisher and ZeitZeiger do not utilize time labels of test samples in the processing of test data, to make a fair comparison of their performance in predicting circadian time, we modified TimeSignatR's code so that the pipeline no longer uses time labels of test samples and it randomly pulls a second sample from the test sample pool to do the 2-point calibration"

The application of TimeSignatR DOES NOT require knowledge of a subject's true circadian time. (It does not require knowing the absolute time labels of the two test samples). Rather it requires collecting two samples separated by a fixed temporal gap. It requires (for example) that a subject has a blood draw now – and again 12 hours later. That can of course be done without have any

knowledge the subject's internal circadian time at either blood draw. For example, imagine a shift worker walks into a clinic and the clinical provider does not know that patient's circadian current circadian phase. They could get a sample now and again in 12 hours. Of course, that requires having the patient return in 12 hours – a very big clinical barrier. But it is not the informatic cheat the authors seem to be implying.

TimeSingatR is effectively premised on the idea that by averaging the expression of a gene now and its expression 12 hours later –one can approximate the baseline/"DC" component of the signal for an individual subject. Similarly the difference between these two samples (the numerator in the time signature scaling) will itself be a sinusoid (without a DC component). Duan et al have themselves concluded that inter-individual human variation in this parameter can have a marked influence on gene expression and classifier performance. TimeSingatR (as I understand it) uses antipodal samples to mitigate this concern.

TimeSingatR is supposed to be trained on data where the lag between blood draws in the training data matches the lag that is used in test data. It simply makes no sense – and is grossly unfair to the method – to use it in the way the authors here suggest. Looking at the sum/difference of samples randomly spaced in time makes no sense. I

The method should be used the way it was developed (use 12 hour lagged data for example in both training and testing). The authors are free to argue (and I would if I were them) that the requirement for 2 blood draws 12 hours apart (or two experimental tissue/cell samples 12 hours apart) is a very significant clinical and/or experimental barrier. But I do not think they are free to use the method as they describe and then disparage it.

Minor:

"CYCLOPS outputs the relative ordering, instead of timestamps, of samples, and requires reconstruction to incorporate every new sample "

This is true, but it also true that unlike your method CYCLOPS does not require training data – it is an unsupervised method (which is also why it can't provide an absolute temporal annotation - only a relative ordering

For fairness would update to:

"CYCLOPS does not require prior training data, but it outputs the relative ordering, instead of timestamps, of samples, and requires reconstruction to incorporate every new sample. "

Reviewer #3 (Remarks to the Author):

The authors have done a thorough review and have addressed all of my questions satisfactorily. I would like to recommend publication in Nature Communications.

Reviewer #1 (Remarks to the Author):

I am satisfied with most answers to my previous questions except for this one regarding Figure 6b-c:

My previous question remains:

"The fact that the mean phase of immune cells is advanced compared to fibroblast at some ZT times but delayed at other ZT times (Figure 6b) indicates that the fluctuation of phases is more due to the estimation procedure itself rather than the biological differences."

We appreciate the reviewer's concern that the difference in the mean and standard deviation (SD) when applied to dermal fibroblasts and immune cells scRNAseq data are caused by the estimation procedure itself instead of biological differences.

There are two main possible sources of the variability we observe: the estimation process and the scRNAseq data that captures biological properties. To make sure that the estimation process itself does not generate significant variability, we performed an experiment. Instead of applying the estimation process on immune cells and fibroblasts, we applied the same estimation process to the same cell type twice, and compared the results. If the estimation process is the source of the variability, then we should observe a significant difference in the prediction outcomes even when the procedure was applied to the same group of cells.

Taking immune cells for example (we performed this with fibroblasts as well and the conclusions were the same), we randomly sampled cells to generate 500 pseudobulk data (Set 1) and repeated the exact same process to generate another 500 pseudobulk data (Set 2). We demonstrate that indeed the two sets of pseudobulk data are random and different:

Round 1 – first 10 of the 500 random pseudobulk

	rep_1	rep_2	rep_3	rep_4	rep_5	rep_6	rep_7	rep_8	rep_9	rep_10
Spata6	40	51	53	53	46	46	37	58	41	31
Alpk1	15	20	16	25	25	14	14	15	14	17
Casp12	4	0	10	5	12	2	1	9	3	4
Pfkfb3	144	165	113	170	138	157	99	165	91	131
Erh	114	80	84	76	96	92	93	60	82	80
Gars	40	33	48	33	34	32	24	25	36	20
Vav2	26	22	18	22	16	21	19	14	28	22
Acly	130	139	106	119	141	132	139	121	95	113
Atg10	31	20	20	13	15	21	19	14	15	21
Mtd1	57	45	67	52	53	40	40	32	52	45
Mrpl23	135	135	114	124	123	142	125	113	122	108
Nr1d1	89	94	76	90	96	84	83	118	84	78
Tbl1xr1	55	48	52	68	59	76	59	57	67	57
Nr1d2	106	104	79	82	96	83	69	92	98	89
Per2	20	36	22	16	20	20	18	11	15	15
Cry1	21	18	13	29	3	14	13	18	23	11

Round 2 – first 10 of the 500 random pseudobulk

	rep_1	rep_2	rep_3	rep_4	rep_5	rep_6	rep_7	rep_8	rep_9	rep_10
Spata6	48	56	43	54	43	50	55	73	45	60
Alpk1	28	14	12	16	19	24	26	8	22	13
Casp12	5	4	3	5	5	5	2	6	6	3
Pfkfb3	155	128	157	148	127	137	163	162	177	141
Erh	77	81	90	67	106	108	83	104	84	119
Gars	31	40	29	30	43	43	39	44	22	40
Vav2	26	12	26	24	34	19	25	16	24	23
Acly	118	116	105	141	104	144	142	132	116	143
Atg10	11	16	15	16	20	15	20	19	14	21
Mtd1	47	51	57	44	56	54	55	60	56	49
Mrpl23	111	121	133	120	117	120	129	127	122	137
Nr1d1	77	79	98	73	94	123	88	98	83	95
Tbl1xr1	50	48	68	53	58	64	59	77	50	69
Nr1d2	80	82	99	88	106	94	88	91	83	87
Per2	26	22	12	20	25	22	20	17	23	23
Cry1	8	13	18	14	20	18	15	11	9	23

We then applied the tauFisher pipeline to make predictions for the pseudobulk samples. The results are plotted in radar plots (below, red = Set 1, blue = Set 2, colors hard to distinguish as they overlap greatly).

We see that the prediction outcome for the two sets greatly overlap with each other. Additionally, no significant differences in mean or SD was found between the two sets:

In summary, the estimation process does not produce significant variability when applied to the same set of data. Since we applied the exact same procedure with the exact same set of parameters to dermal fibroblast and immune cells, the scRNAseq data that captures biological properties is the source of the variability we observed and reported in the manuscript, not the estimation process.

In the revised paper, the authors stated that "Whether one cell type's circadian clock is ahead of the other is inconclusive (Figure 6c)." but statistically significant differences (p -value ≤ 0.0001) were found for most ZT times. ZT6 even has the opposite sign from other ZTs.

We appreciate the reviewer's comment regarding the statistically significant differences in prediction means of the fibroblasts and the immune cells. In an ideal example with minimum noise, where immune cells' internal clocks are ahead of the fibroblast clocks by 3 hours, we

should see prediction mean to differ by 3 no matter when we sample the cells: fibroblasts at ZT0 while immune cells at ZT3, fibroblasts at ZT2 while immune cells at ZT5, fibroblasts at ZT10 while immune cells at ZT13 and so on. Since the prediction outcome suggests that immune cells are ahead of the fibroblasts at some time points (ZT18, 20) and the fibroblasts are ahead of the immune cells at the others (ZT2, 6, 14), we wrote "Whether one cell type's circadian clock is ahead of the other is inconclusive (Figure 6c)."

Furthermore, the means of predicted time stamps of fibroblasts and immune cells deviate from real sample collection times at several ZTs, e.g. ZT10. This seems contradictory to the robustness of phase estimation claimed in the text.

In Figure 6, the training data are the whole mouse skin microarray data, whereas the test datasets are randomly sampled pseudobulk replicates obtained from immune cells only or fibroblasts only. Figure 3g better demonstrates tauFisher's robustness in adding time labels to pseudobulk data calculated from the entire samples (accuracy 0.778, RMSE 2.198) that contains not only fibroblasts and immune cells, but also muscle cells and endothelial cells that can contribute to circadian time prediction outcome if included in the pseudobulk data.

Additionally, we want to emphasize that it is not necessarily true that fibroblasts and immune cells collected *in vivo* possess circadian clocks that are synchronized with the external time labels (i.e. Fibroblasts could be at ZT13 even though the external time label is ZT10). Previous multiorgan studies using bulk level expression data have found that clock genes show phase-shifted patterns when compared across different organs (Zhang et al., 2014). Current research has not eliminated the possibility that such phase shifts may also occur at a cell type level, and we attempted to gain insights into this question, but this effort was met with inconclusive results ("Whether one cell type's circadian clock is ahead of the other is inconclusive (Figure 6c)."). We plan to delve further into this question in our future scRNAseq experiments.

In summary, due to the possible phase shifts at cell type level, circadian phases of randomly sampled pseudobulk datasets of fibroblasts and immune cells are not necessarily aligned with the external time label. Results in Figure 2 and 3 where tauFisher was trained and tested using whole samples in different tissues, assay platforms, and cross-platforms scenarios are more useful for assessing robustness.

The authors continued on to say "Indeed, the standard deviation of the prediction distribution is significantly greater for immune cells for five out of the six ZTs." but again statistically significant differences were found for standard deviation for most ZT times. The difference of SD at ZT6 is again opposite to the differences at other ZTs.

We agree with the reviewer that statistically significant differences were found in SD for most ZT times and for five out of the six ZTs it is the immune cells that have higher SD than fibroblast. Therefore, we wrote "Indeed, the standard deviation of the prediction distribution is significantly greater for immune cells for five out of the six ZTs.". The prediction outcome at ZT6 for fibroblast is very interesting as the distribution appears to be bimodal whereas at the other 5 ZTs the

distributions are relatively unimodal. Since the SD is greater in immune cells than in fibroblasts for 5 out of the 6 time points, we wrote:

“We hypothesize that the circadian clock is more heterogeneous in dermal immune cells than in dermal fibroblasts, and such heterogeneity may be the reason behind the dampened core clock and fewer rhythmic genes we found in immune cells based on collective, cell-type level, gene expression data. Such a result is not unexpected, as the fibroblasts may be more homogeneous in their biological function than the immune cells, which contain dendritic cells as well as different types of macrophages and lymphocytes (Supplementary Figure 6) that serve different immune functions. Unfortunately, we did not capture enough cells for each specific immune cell types in the scRNAseq experiment to generate reliable pseudobulk data that is required for further circadian analysis (Supplementary Figure 6).”

I also suggest the authors change the wordings to avoid over-statement in places such as:

"unprecedentedly, tauFisher trained on bulk sequencing data is able to accurately predict the circadian time of single-cell RNA sequencing (scRNAseq) pseudobulk data"

"However, tauFisher is the only method that successfully predicted the circadian time for the pseudobulk data"

We thank the reviewer for this suggestion and have gone through the entire manuscript carefully to change the wordings. For the two sentences above, we rewrote them to be

“tauFisher trained on bulk sequencing data is able to accurately predict the circadian time of single-cell RNA sequencing (scRNAseq) pseudobulk data.”

and

“tauFisher outperformed ZeitZeiger and the two-sample TimeSignatR method, and ...”

Reviewer #2 (Remarks to the Author):

I thank the Authors for taking the time to try to thoroughly address my concerns. They have improved the paper.

However, as they clarified their benchmarking methods, it is now clear they have been quite unfair in the benchmarking/analysis of the TimeSignatR method. This is now (unfortunately) a major concern.

Major:

The text added to the manuscript (in response to my question about TimeSignatR) is given below:

“The main cause can be attributed to our modification of their 2-point calibration step (Methods). In the original TimeSignatR pipeline, time labels of test samples are available to TimeSignatR as it needs to pick out antipodal test samples that are around 12 hours apart in the calibration step (Supplementary File). As tauFisher and ZeitZeiger do not utilize time labels of test samples in the processing of test data, to make a fair comparison of their performance in predicting circadian time, we modified TimeSignatR’s code so that the pipeline no longer uses time labels of test samples and it randomly pulls a second sample from the test sample pool to do the 2-point calibration”

The application of TimeSignatR DOES NOT require knowledge of a subject’s true circadian time. (It does not require knowing the absolute time labels of the two test samples). Rather it requires collecting two samples separated by a fixed temporal gap. It requires (for example) that a subject has a blood draw now – and again 12 hours later. That can of course be done without have any knowledge the subject’s internal circadian time at either blood draw. For example, imagine a shift worker walks into a clinic and the clinical provider does not know that patient’s circadian current circadian phase. They could get a sample now and again in 12 hours. Of course, that requires having the patient return in 12 hours – a very big clinical barrier. But it is not the informatic cheat the authors seem to be implying.

TimeSignatR is effectively premised on the idea that by averaging the expression of a gene now and its expression 12 hours later –one can approximate the baseline/“DC” component of the signal for an individual subject. Similarly the difference between these two samples (the numerator in the time signature scaling) will itself be a sinusoid (without a DC component). Duan et al have themselves concluded that inter-individual human variation in this parameter can have a marked influence on gene expression and classifier performance. TimeSingatR (as I understand it) uses antipodal samples to mitigate this concern.

TimeSingatR is supposed to be trained on data where the lag between blood draws in the training data matches the lag that is used in test data. It simply makes no sense – and is grossly unfair to the method – to use it in the way the authors here suggest. Looking at the sum/difference of samples randomly spaced in time makes no sense. I

The method should be used the way it was developed (use 12 hour lagged data for example in both training and testing). The authors are free to argue (and I would if I were them) that the requirement for 2 blood draws 12 hours apart (or two experimental tissue/cell samples 12 hours apart) t is a very significant clinical and/or experimental barrier. But I do not think they are free to use the method as they describe and then disparage it.

We have now implemented TimeSignatR the way it was developed and provided the pipeline paired test samples that are 12 hours apart. TimeSignatR showed greatly improved performance in benchmarking and we have updated Figure 2 and 3 accordingly.

We have updated the related sentences in the Introduction:

“But, they have limitations. ... TimeSignatR requires two test samples and it achieves its best performance when the two samples are 12 hours apart.”

We have also updated the Result section, highlighting that TimeSignatR's two-sample within-subject normalization effectively addresses human individual variability in circadian phase estimation:

For eleven out of the twelve benchmarking datasets, tauFisher achieved higher accuracy when using predictor genes found by JTK Cycle [32] instead of Lomb-Scargle [33]. For six out of the ten transcriptomic datasets collected from mice, tauFisher achieved equal or higher 2-hour accuracy using one test sample than TimeSignatR using two test samples that are 12 hours apart. tauFisher achieved lower but comparable accuracy (difference < 10%) when compared to TimeSignatR in two of the remaining four mouse datasets. For the two human blood datasets, TimeSignatR using two test samples outperformed ZeitZeiger and tauFisher, highlighting the importance and effectiveness of using two test samples to address human variability in circadian phase predictions. ZeitZeiger could not predict the time for several iterations due to linearly dependent basis vectors. Interestingly, whether ZeitZeiger ran into linear dependency issues appeared to be depended on the assay methods, as it ran successfully for most of the microarray data but failed to predict the time for all 100 iterations in the bulk RNAseq datasets collected from mouse kidney, liver, brainstem, and cerebellum.

A method called TimeMachine was published during this round of revision and we updated the Discussion section:

“Despite its unique circadian time prediction ability, ... living style and diet. A recently proposed pipeline called TimeMachine attempted to infer circadian phase from a single human blood sample and reported 2-hour accuracy of 40~55% [45]. On the other hand, TimeSignatR using two human blood samples 12 hours apart achieved ~73% 2-hour accuracy (Figure 2), emphasizing the necessity to address individual variability when predicting circadian phases for human samples and the effectiveness of the two-sample within-subject normalization step. One assumption...”

Minor:

“CYCLOPS outputs the relative ordering, instead of timestamps, of samples, and requires reconstruction to incorporate every new sample “

This is true, but it also true that unlike your method CYCLOPS does not require training data – it is an unsupervised method (which is also why it can't provide an absolute temporal annotation - only a relative ordering

For fairness would update to:

“CYCLOPS does not require prior training data, but it outputs the relative ordering, instead of timestamps, of samples, and requires reconstruction to incorporate every new sample.”

We thank the reviewer for the thorough consideration. We have changed the sentence to

“CYCLOPS outputs the relative ordering, instead of timestamps, of samples, and requires reconstruction to incorporate every new sample as it does not require prior training data.”

Reviewer #3 (Remarks to the Author):

The authors have done a thorough review and have addressed all of my questions satisfactorily. I would like to recommend publication in Nature Communications.

We appreciate the constructive feedback and are glad to hear that we have addressed all your questions satisfactorily. Thank you for your valuable comments and inputs.

Reference:

Zhang, R., Lahens, N. F., Ballance, H. I., Hughes, M. E., & Hogenesch, J. B. (2014). A circadian gene expression atlas in mammals: Implications for biology and medicine. *Proceedings of the National Academy of Sciences*, 111(45), 16219–16224.
<https://doi.org/10.1073/pnas.1408886111>

Reviewer #1 (Remarks to the Author):

I am fine with authors' replies now.

Reviewer #2 (Remarks to the Author):

The authors have addressed my concerns